# TOOLGEN: UNIFIED TOOL RETRIEVAL AND CALLING VIA GENERATION

**Renxi Wang**[1,2]      **Xudong Han**[1,2]      **Lei Ji**[3]      **Shu Wang**[4]
**Timothy Baldwin**[1,2,5]      **Haonan Li**[1,2]
[1]LibrAI     [2]Mohamed bin Zayed University of Artificial Intelligence     [3]Microsoft
[4]University of California, Los Angeles     [5]The University of Melbourne
{renxi.wang,xudong.han,timothy.baldwin,haonan.li}@mbzuai.ac.ae
leiji@microsoft.com      shuwang0712@ucla.edu

## ABSTRACT

As large language models (LLMs) advance, their inability to autonomously execute tasks by directly interacting with external tools remains a critical limitation. Traditional methods rely on inputting tool descriptions as context, which is constrained by context length and requires separate, often inefficient, retrieval mechanisms. We introduce ToolGen, a paradigm shift that integrates tool knowledge directly into the LLM's parameters by representing each tool as a unique token. This enables the LLM to generate tool calls and arguments as part of its next token prediction capabilities, seamlessly blending tool invocation with language generation. Our framework allows the LLM to access and utilize a vast amount of tools with no additional retrieval step, significantly enhancing both performance and scalability. Experimental results with over 47,000 tools show that ToolGen not only achieves superior results in both tool retrieval and autonomous task completion but also sets the stage for a new era of AI agents that can adapt to tools across diverse domains. By fundamentally transforming tool retrieval into a generative process, ToolGen paves the way for more versatile, efficient, and autonomous AI systems. ToolGen enables end-to-end tool learning and opens opportunities for integration with other advanced techniques such as chain-of-thought and reinforcement learning, thereby expanding the practical capabilities of LLMs[1].

## 1 INTRODUCTION

Large language models (LLMs) have demonstrated impressive capabilities as interactive systems, adept at processing external inputs, executing actions, and autonomously completing tasks (Gravitas, 2023; Qin et al., 2023; Yao et al., 2023; Shinn et al., 2023; Wu et al., 2024a; Liu et al., 2024; Wang et al., 2024b;c). Among the various methods enabling LLMs to interact with the world, tool calling via APIs has emerged as one of the most common and effective approaches. However, as the number of tools grows into the tens of thousands, existing methods for tool retrieval and execution struggle to scale efficiently.

A common approach in real-world scenarios is to combine tool retrieval with tool execution, where a retrieval model first narrows down the relevant tools before passing them to the LLM for final selection and execution (Qin et al., 2023; Patil et al., 2023). While this combined method addresses the challenge of handling vast numbers of tools, it has notable limitations: retrieval models often rely on small encoders that fail to fully capture the semantics of complex tools and queries, and separating retrieval from execution introduces inefficiencies and potential misalignment between stages of task completion.

Moreover, LLMs and their tokenizers are pretrained primarily on natural language data (Brown et al., 2020; Touvron et al., 2023), leaving them with limited intrinsic knowledge of tool-related functionalities. This gap in knowledge results in suboptimal performance, especially when the LLM must rely on retrieved tool descriptions for decision-making.

---

[1]Data and code are available at `https://github.com/Reason-Wang/ToolGen`

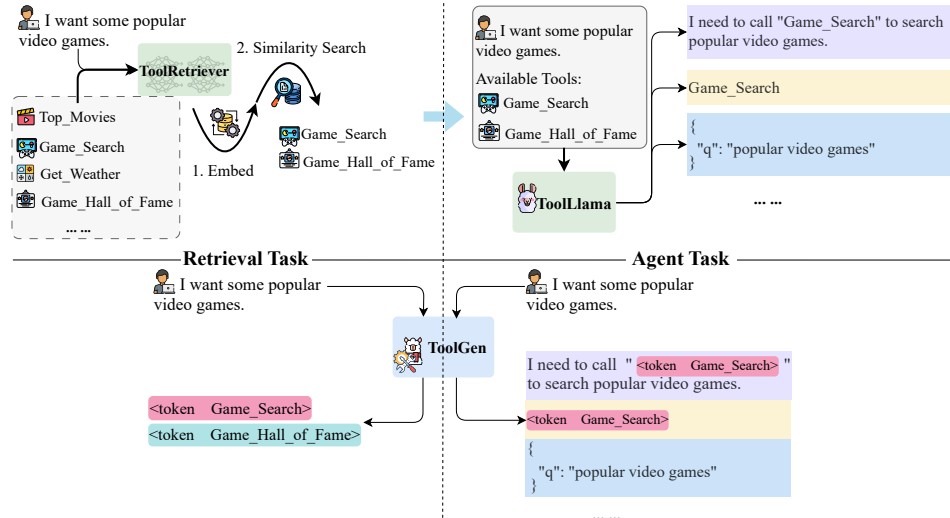

Figure 1: Comparison between previous retrieval-based methods and our ToolGen. Previous methods use a retriever to retrieve relevant tools based on similarity matching, which are further put into prompts for LLMs to select. ToolGen can retrieve tools by generating tool tokens directly. ToolGen can also complete the task without relying on any external retriever.

In this study, we introduce **ToolGen**, a novel framework that integrates real-world tool knowledge directly into the LLM's parameters and transforms tool retrieval and execution into a unified generation task. Specifically, ToolGen expands the LLM's vocabulary with tool-specific virtual tokens and trains the model to generate these tokens within a conversational context, allowing the LLM to leverage its pre-existing knowledge more effectively for both retrieving and calling tools.

Specifically, each tool is represented as a unique virtual token within the LLM's vocabulary. Building upon a pretrained LLM, ToolGen's training process consists of three stages: tool memorization, retrieval training, and agent training. In the tool memorization stage, the model associates each virtual tool token with its documentation. During retrieval training, the model learns to generate relevant tool tokens based on user queries. Finally, in end-to-end agent-tuning, the model is trained to act as an autonomous agent, generating plans and tools, and determining the appropriate parameters to complete tasks. By calling tools and receiving feedback from external environments, the model can handle user queries efficiently and integratively. Figure 1 shows comparison between ToolGen and traditional paradigms.

We demonstrate ToolGen's superiority in two scenarios: a tool retrieval task, where the model retrieves the correct tool for a given query, and an LLM-based agent task, where the model completes complex tasks involving real-world API calls. Leveraging a dataset of 47,000 real-world tools, ToolGen achieves performance comparable to the leading tool retrieval methods, but with significantly lower cost and greater efficiency. Additionally, it surpasses traditional tool learning paradigms, highlighting its potential for advancing more effective tool usage systems.

ToolGen represents a paradigm shift in tool interaction by merging retrieval and generation into a single, cohesive model. This innovation sets the stage for a new generation of AI agents capable of adapting to a vast array of tools across diverse domains. Additionally, ToolGen opens new opportunities for integrating advanced techniques like chain-of-thought reasoning and reinforcement learning with the ability to use tools in a unified generation way, expanding the capabilities of LLMs in real-world applications.

In summary, our contributions are:

- A novel framework, ToolGen, that integrates tool retrieval and execution into the LLM's generative process using virtual tokens.

- A three-stage training process that enables efficient and scalable tool retrieval and API calling within ToolGen.

- Experimental validation demonstrates that ToolGen achieves comparable performance to current best tool retrieval methods with significantly less cost and higher efficiency and surpasses traditional tool learning paradigms across large-scale tool repositories.

## 2 RELATED WORK

### 2.1 TOOL RETRIEVAL

Tool retrieval is essential for LLM agents in real-world task execution, where tools are usually represented by their documentation. Traditional methods like sparse (e.g., BM25 (Robertson et al., 2009)) and dense retrieval (e.g., DPR (Karpukhin et al., 2020), ANCE (Xiong et al., 2021)) rely on large document indices and external modules, leading to inefficiencies and difficulty in optimizing in an end-to-end agent framework. Some work has explored alternative methods. For example, Chen et al. (2024b) rewrite queries and extract their intent, targeting unsupervised retrieval settings, though the results are not comparable to supervised approaches. Xu et al. (2024) propose a method that iteratively refines queries based on tool feedback, improving retrieval accuracy but increasing latency.

Recently, generative retrieval has emerged as a promising new paradigm, wherein models directly generate relevant document identifiers rather than relying on traditional retrieval mechanisms (Wang et al., 2022; Sun et al., 2023b; Kishore et al., 2023; Mehta et al., 2023b; Chen et al., 2023c). Motivated by this, ToolGen represents each tool as a unique token, allowing tool retrieval and calling to be framed as a generation task. Beyond simplifying retrieval, this design integrates smoothly with other LLM and LLM-based agent features like chain-of-thought reasoning (Wei et al., 2023) and ReAct (Yao et al., 2023). By consolidating retrieval and task execution into a single LLM agent, it reduces latency and computational overhead, leading to more efficient and effective task completion.

### 2.2 LLM-AGENTS WITH TOOL CALLING

LLMs have shown strong potential in mastering tools for various tasks. However, most existing works focus on a limited set of actions (Chen et al., 2023a; Zeng et al., 2023; Yin et al., 2024; Wang et al., 2024a). For instance, Toolformer (Schick et al., 2023) fine-tunes GPT-J to handle just five tools, such as calculators. While effective for narrow tasks, this approach struggles in real-world scenarios with vast action spaces. ToolBench (Qin et al., 2023) expands the scope by introducing over 16,000 tools, highlighting the challenge of tool selection in complex environments.

To perform tool selection, current methods often use a retriever-generator pipeline, where relevant tools are retrieved and then utilized by the LLM (Patil et al., 2023; Qin et al., 2023). In addition, TPTU (Ruan et al.) proposes a structured framework for LLM agents and evaluates their task planning and tool usage abilities. Furthermore, TPTU-v2 (Kong et al.; 2024) builds an LLM Finetuner to enhance agent performance with curated datasets and a demo selector to select relevant demonstrations. They set a flexible and superior paradigm compared to traditional retrieval-based paradigm. However, pipelined approaches face two major issues: error propagation from the retrieval step and the inability of LLMs to fully understand and use tools via simple prompting.

To mitigate these issues, researchers have tried representing actions as tokens, converting action prediction into a generative task. For example, RT2 (Brohan et al., 2023) generates tokens representing robot actions, and Self-RAG (Asai et al., 2023) uses special tokens to decide when to retrieve documents. ToolkenGPT (Hao et al., 2023) introduces tool-specific tokens to trigger tool usage, a concept closest to our approach.

Our approach differs from ToolkenGPT in several ways. First, we focus on real-world tools that require flexible parameters for complex tasks (e.g., YouTube channel search), while ToolkenGPT is limited to simpler tools with fewer inputs (e.g., math functions with two numbers). Additionally, ToolkenGPT relies on few-shot prompting, whereas ToolGen incorporates tool knowledge directly into the LLM through full-parameter fine-tuning, enabling the model to retrieve and execute tasks autonomously. Finally, our experiments involve a much larger tool set—47,000 tools compared to ToolkenGPT's 13–300. Detailed comparison and other related work can be found in Section A.

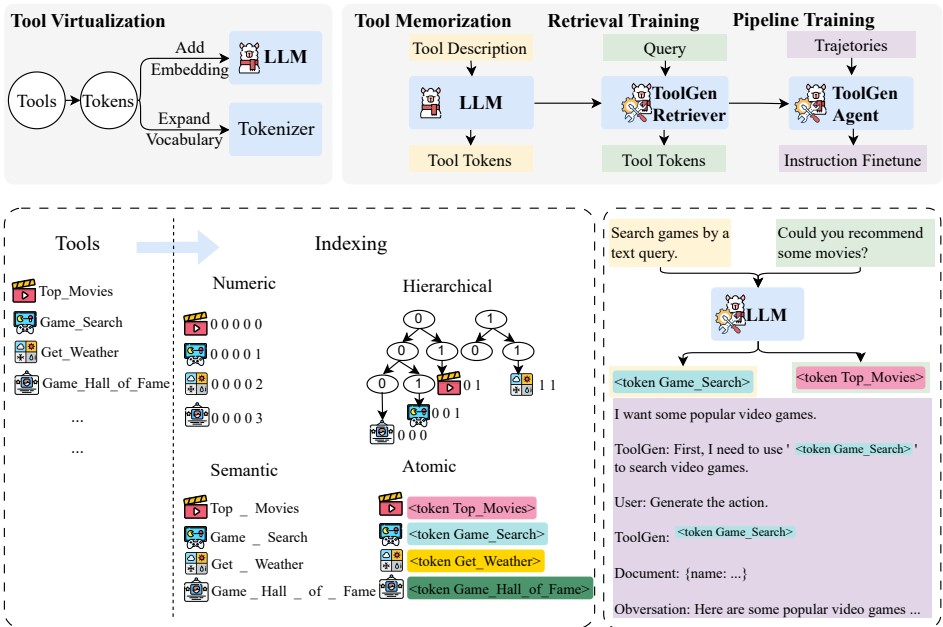

Figure 2: An illustration of ToolGen framework. In tool virtualization, tools are mapped into virtual tokens. In the following three-stage training, ToolGen first memorizes tools by predicting tool tokens based on their documentations. Then it learns to retrieve tools by predicting tool tokens from queries. Finally, pipeline data, i.e., trajectories, are used to finetune the retriever model from the last stage, resulting in the ToolGen Agent model.

## 3 TOOLGEN

In this section, we first introduce the notations used throughout the paper. Then we detail the specific methods of ToolGen, including tool virtualization, tool memorization, retrieval training, and end-to-end agent tuning, as illustrated in Figure 2. Lastly, we describe our inference approach.

### 3.1 PRELIMINARIES

Given a user query $q$, tool learning aims to resolve $q$ using tools from a large tool set $D = \{d_1, d_2, \ldots, d_N\}$, where $|D| = N$ is a large number, making it impractical to include all tools in $D$ in the LLM context. Therefore, current research typically uses a retriever $R$ to retrieve $k$ relevant tools from $D$, denoted as $D_{k,R} = \{d_{r_1}, d_{r_2}, \ldots, d_{r_k}\} = R(q, k, D)$, where $|D_{k,R}| \ll N$. The final prompt is then the concatenation of $q$ and $D_{k,R}$, denoted as $Prompt = [q, D_{k,R}]$. To complete a task (query), an LLM-based agent usually adopts a four-stage paradigm (Qu et al., 2024) iteratively: generates a plan $p_i$, selects a tool $d_{si}$, determines tool parameters $c_i$, and collects feedback from the tool(s) $f_i$. We denote these steps for the $i$-th iteration as $p_i, d_{s_i}, c_i, f_i$. The model continues iterating through these steps until the task is completed, at which point the final answer $a$ is generated. The entire trajectory can be represented as $Traj = [Prompt, (p_1, d_{s_1}, c_1, f_1), \ldots, (p_t, d_{s_t}, c_t, f_t), a] = [q, R(q, D), (p_1, d_{s_1}, c_1, f_1), \ldots, (p_t, d_{s_t}, c_t, f_t), a]$. This iterative approach allows the model to dynamically adjust and refine its actions at each step based on the feedback received, improving its performance in completing complex tasks.

### 3.2 TOOL VIRTUALIZATION

In ToolGen, we virtualize tools by mapping each tool to a unique new token through a method we call atomic indexing. In this approach, each tool is assigned a unique token by expanding the LLM's vocabulary. The embedding for each tool token is initialized as the average embedding of its corresponding tool name, ensuring a semantically meaningful starting point for each tool.

Formally, the token set is defined as $T = \text{Index}(d) \mid \forall d \in D$, where $\text{Index}$ is the function mapping tools to tokens. We demonstrate that atomic indexing is more efficient and can mitigate hallucination compared to other indexing methods, such as semantic and numeric mappings, discussed in Section 4.3 and 5.4.

### 3.3 Tool Memorization

After assigning tokens to tools, the LLM still lacks any knowledge of the tools. To address this, we inject tool information by fine-tuning it with tool descriptions as inputs and their corresponding tokens as outputs, which we call tool memorization. We use the following loss function:

$$\mathcal{L}_{tool} = \sum_{d \in D} - \log p_\theta(\text{Index}(d)|d_{doc})$$

where $\theta$ denotes the LLM parameters, and $d_{doc}$ represents the tool description. This step equips the LLM with basic knowledge of the tools and their associated actions.

### 3.4 Retrieval Training

We then train the LLMs to link the hidden space of virtual tool token (and its documentation), to the user query space, so that LLM can generate correct tool based on a user's query. To achieve this, we fine-tune the LLM with user queries as inputs and corresponding tool tokens as outputs:

$$\mathcal{L}_{retrieval} = \sum_{q \in Q} \sum_{d \in D_q} - \log p_{\theta'}(\text{Index}(d)|q)$$

where $\theta'$ represents the LLM parameters after tool memorization, $Q$ is the set of user queries, and $D_q$ is the set of tools relevant to each query. This results in the ToolGen Retriever, which can generate the appropriate tool token given a user query.

### 3.5 End-to-End Agent-Tuning

After retrieval training, the LLM is capable of generating tool tokens from queries. In the final stage, we fine-tune the model with agent task completion trajectories. We adopt a similar inference strategy as Agent-Flan (Chen et al., 2024c), in instead of generating Thought, Action, and Arguments together as `ReAct`. Our pipeline follows an iterative process, where the LLM first generates a Thought, and the corresponding Action token. This token is used to fetch the tool documentation, which the LLM uses to generate the necessary arguments. The process continues iteratively until the model generates a "finish" token or the maximum number of turns is reached. The generated trajectory is represented as $Traj = [q, (p_1, \text{Index}(d_{s_1}), c_1, f_1), \ldots, (p_t, \text{Index}(d_{s_t}), c_t, f_t), a]$. In this structure, relevant tools are no longer required.

### 3.6 Inference

During inference, the LLM may generate action tokens outside the predefined tool token set. To prevent this, we designed a constrained beam search generation that restricts the output tokens to the tool token set. We applied this constrained beam search for both tool retrieval, where the model selects tools based on queries, and the end-to-end agent system, significantly reducing hallucination during the action generation step. A detailed analysis can be found in Section 5.4. The implementation details can be found in Appendix E.

## 4 Tool Retrieval Evaluation

### 4.1 Experimental Setup

We use pretrained Llama-3-8B (Dubey et al., 2024) as our foundation model, with a vocabulary size of 128,256. Using the atomic indexing approach, we expand the vocabulary by an additional 46,985 tokens following the tool virtualization process, resulting in a final vocabulary size of 175,241. We fine-tune the model using the Llama-3 chat template with a cosine learning rate scheduler, applying

a 3% warm-up steps. The maximum learning is $4 \times 10^{-5}$. All models are trained using Deepspeed ZeRO 3 (Rajbhandari et al., 2020) across 4×A100 GPUs. We train 8 epochs for tool memorization and 1 epoch for retrieval training.

**Dataset**  Our experiments are based on ToolBench, a real-world tool benchmark containing more 16k tool collections, each containing several APIs, resulting in a total of 47k unique APIs. Each API is documented with a dictionary, containing the name, description, and parameters for calling the API. A real example is shown in Appendix C. We take each API as an action and map it to a token. Our retrieval and end-to-end agent-tuning data are converted from the original data in ToolBench. Details can be found in Appendix K. Although each tool may consist of multiple APIs, for simplicity, we refer to each API as a tool in this paper.

We follow the data split of Qin et al. (2023), where 200k (query, relevant API) pairs are divided into three categories: I1 (single-tool queries), I2 (intra-category multi-tool queries), and I3 (intra-collection multi-tool instructions), containing 87,413, 84,815, and 25,251 instances, respectively.

**Baselines**  We compare ToolGen with the following baselines:

- BM25: A classical unsupervised retrieval method based on TF-IDF, which retrieves documents based on term similarity with the query.
- Long-Context LLMs: We concatenate tools into a long prompt to `gpt-4o`, and prompt it to choose from the pool. Limit by context length, we cannot input all 47k tools, so we use 2k tools with ground truth tools included.
- Embedding Similarity (EmbSim): Sentence embeddings generated using OpenAI's sentence embedding model; specifically `text-embedding-3-large` in our experiences.
- Re-Invoke (Chen et al., 2024b): An unsupervised retrieval method with query rewriting and document expansion.
- IterFeedback (Xu et al., 2024): BERT-based retriever with `gpt-3.5-turbo-0125` as a feedback model with iterative feedback for up to 10 rounds.
- ToolRetriever (Qin et al., 2023): A BERT-based retriever trained via contrastive learning.

**Settings**  We conduct experiments under two settings. In the first, **In-Domain Retrieval**, the retrieval search space is restricted to tools within the same domain. For example, when evaluating queries from domain I1, the search is limited to I1 tools. This aligns with ToolBench settings. The second, **Multi-Domain Retrieval**, is more complex, with the search space expanded to include tools from all three domains. In this case, models are trained on combined data, increasing both the search space and task complexity. Unlike ToolBench, this multi-domain setting reflects real-world scenarios where retrieval tasks may involve overlapping or mixed domains. This setup evaluates the model's ability to generalize across domains and handle more diverse, complex retrieval cases.

**Metrics**  We evaluate retrieval performance using Normalized Discounted Cumulative Gain (NDCG) (Järvelin & Kekäläinen, 2002), a widely used metric in ranking tasks, including tool retrieval. NDCG accounts for both the relevance and ranking position of retrieved tools.

### 4.2 RESULTS

Table 1 presents the tool retrieval results. As expected, all trained models significantly outperform the untrained baselines (BM25, EmbSim, and Re-Invoke) across all metrics, demonstrating the benefit of training on tool retrieval data.

Our proposed ToolGen model consistently achieves the best performance across both settings. In the In-Domain setting, ToolGen delivers highly competitive results, achieving comparable performance to the IterFeedback system, which uses multiple models and a feedback mechanism. ToolGen, as a single model, outperforms ToolRetriever by a significant margin in all metrics and even surpasses IterFeedback in several cases, such as NDCG@5 for domain I1 and NDCG@1,@3,@5 for I2.

In the Multi-Domain setting, where the search space is larger and performance generally drops, ToolGen remains robust, outperforming ToolRetriever and maintaining superiority over other baselines. This demonstrates that ToolGen, despite being a single model, is capable of competing with

Table 1: Tool retrieval evaluation across two settings: (1) **In-Domain**, where models are trained and evaluated within the same domain; and (2) **Multi-Domain**, where models are trained on all domains and evaluated with the full set of tools across all domains. BM25, EmbSim, and Re-Invoke are unsupervised baselines without training. IterFeedback is retrieval system with multiple models and feedback mechanism. ToolRetriever is trained using contrastive learning, while ToolGen is trained with next-token prediction. Results marked with * were not implemented by us and are copied from their original paper, and hence only in the In-Domain setting. For ToolGen in the In-Domain setting, we allow the generation space to include all tokens, which is a more challenging scenario compared to other models. Best results in each category are **bolded**.

| Model | I1 | | | I2 | | | I3 | | |
|---|---|---|---|---|---|---|---|---|---|
| | NDCG1 | NDCG3 | NDCG5 | NDCG1 | NDCG3 | NDCG5 | NDCG1 | NDCG3 | NDCG5 |
| **In-Domain** | | | | | | | | | |
| BM25 | 29.46 | 31.12 | 33.27 | 24.13 | 25.29 | 27.65 | 32.00 | 25.88 | 29.78 |
| Long-Context LLM* | 32.22 | 42.87 | 52.14 | 25.39 | 33.91 | 46.07 | 25.11 | 32.57 | 44.03 |
| EmbSim | 63.67 | 61.03 | 65.37 | 49.11 | 42.27 | 46.56 | 53.00 | 46.40 | 52.73 |
| Re-Invoke* | 69.47 | – | 61.10 | 54.56 | – | 53.79 | 59.65 | – | 59.55 |
| IterFeedback* | **90.70** | **90.95** | 92.47 | 89.01 | 85.46 | 87.10 | **91.74** | **87.94** | **90.20** |
| ToolRetriever | 80.50 | 79.55 | 84.39 | 71.18 | 64.81 | 70.35 | 70.00 | 60.44 | 64.70 |
| ToolGen | 89.17 | 90.85 | **92.67** | **91.45** | **88.79** | **91.13** | 87.00 | 85.59 | 90.16 |
| **Multi-Domain** | | | | | | | | | |
| BM25 | 22.77 | 22.64 | 25.61 | 18.29 | 20.74 | 22.18 | 10.00 | 10.08 | 12.33 |
| EmbSim | 54.00 | 50.82 | 55.86 | 40.84 | 36.67 | 39.55 | 18.00 | 17.77 | 20.70 |
| ToolRetriever | 72.31 | 70.30 | 74.99 | 64.54 | 57.91 | 63.61 | 52.00 | 39.89 | 42.92 |
| ToolGen | **87.67** | **88.84** | **91.54** | **83.46** | **86.24** | **88.84** | **79.00** | **79.80** | **84.79** |

complex retrieval systems like IterFeedback, showcasing its ability to handle complex real-world retrieval tasks where domain boundaries are less defined.

## 4.3 INDEXING METHOD COMPARISON

While ToolGen uses atomic indexing for tool virtualization, we explore several alternative generative retrieval approaches. In this section, we compare it with the following three methods:

- **Numeric**: Map each tool to a unique number. The resulting token is purely numeric, offering no inherent semantic information, but providing a distinct identifier for each tool.

- **Hierarchical**: This method clusters tools into non-overlapping groups and recursively partitions these clusters, forming a hierarchical structure. The index from the root to the leaf in this tree-like structure represents each tool, similarly to Brown clustering techniques.

- **Semantic**: In this approach, each tool is mapped to its name, using the semantic content of the tool names to guide the LLM. The tool's name provides a meaningful representation directly related to its function.

The implementation details are described in Appendix D.

First, we conducted an analysis of the number of subtokens required to represent each tool for the different methods, as shown in Figure 3. Atomic indexing ensures each tool to be a single token, while numeric indexing encodes tools into N tokens for tools numbered in $(10^{N-1}, 10^N]$. In contrast, both semantic indexing and hierarchical indexing produce a variable number of subtokens, with semantic indexing having more outliers with significantly longer sequences. The figure highlights the superiority of atomic indexing, where each

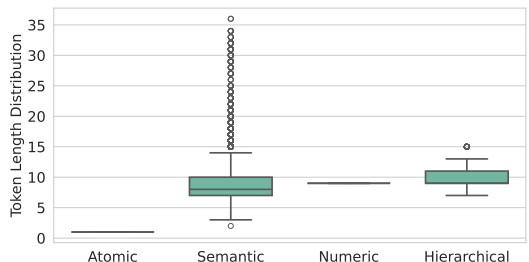

Figure 3: The distribution of the number of subtokens per tool varies across different indexing methods.

tool is represented by a single token, whereas other methods require multiple tokens. This efficiency allows ToolGen to reduce the number of generation tokens and inference time in both the retrieval and agent scenarios.

Table 2: Retrieval evaluation for different indexing methods in Multi-Domain setting. Best results are **bolded** and second best results are underlined.

| Model | I1 | | | I2 | | | I3 | | |
|---|---|---|---|---|---|---|---|---|---|
| | NDCG1 | NDCG3 | NDCG5 | NDCG1 | NDCG3 | NDCG5 | NDCG1 | NDCG3 | NDCG5 |
| Numetric | 83.17 | 84.99 | 88.73 | 79.20 | 79.23 | 83.88 | 71.00 | 74.81 | 82.95 |
| Hierarchical | 85.67 | 87.38 | 90.26 | 82.22 | 82.70 | 86.63 | 78.50 | 79.47 | 84.15 |
| Semantic | **89.17** | **91.29** | **93.29** | **83.71** | 84.51 | 88.22 | **82.00** | 78.86 | **85.43** |
| Atomic | 87.67 | 88.84 | 91.54 | 83.46 | **86.24** | **88.84** | 79.00 | **79.80** | 84.79 |

Table 3: Ablation study for tool retrieval. We assess the impact of removing retrieval training, tool memorization, and constrained beam search on ToolGen's performance, respectively.

| Model | I1 | | | I2 | | | I3 | | |
|---|---|---|---|---|---|---|---|---|---|
| | NDCG1 | NDCG3 | NDCG5 | NDCG1 | NDCG3 | NDCG5 | NDCG1 | NDCG3 | NDCG5 |
| ToolGen | 87.67 | 88.84 | 91.54 | 83.46 | 86.24 | 88.84 | 79.00 | 79.80 | 84.79 |
| −memorization | 84.00 | 86.77 | 89.35 | 82.21 | 83.20 | 86.78 | 77.00 | 77.71 | 84.37 |
| −retrieval training | 10.17 | 12.31 | 13.89 | 5.52 | 7.01 | 7.81 | 3.00 | 4.00 | 4.43 |
| −constraining | 87.67 | 88.79 | 91.45 | 83.46 | 86.24 | 88.83 | 79.00 | 79.93 | 84.92 |

Next, we examined the effectiveness of different indexing methods. As shown in Table 2, semantic indexing demonstrates the best retrieval performance across various metrics and scenarios, while atomic indexing closely follows in many cases. We attribute this to the fact that semantic indexing aligns better with the pretraining data of LLMs. However, this advantage diminishes as the training data and type increase. For example, in Section 5.3, we show that atomic indexing achieves better end-to-end results. We also show that combining constrained beam search with semantic indexing will cause biased tool usage, which is detailed in Section E.2.

Taking all these factors into account, we choose atomic indexing for ToolGen tool virtualization.

### 4.4 ABLATION

We perform an ablation study to assess the impact of different training stages of ToolGen, as shown in Table 3. The results indicate that retrieval training is the crucial factor for tool retrieval performance, as it directly aligns with the retrieval task where inputs are queries and outputs are tool tokens. Removing tool memorization leads to a minor performance drop although it plays a role in improving generalization, which we will discuss further in Appendix J. Similarly, constrained beam search, while not a major contributor to retrieval task, helps prevent hallucinations, making it useful for end-to-end agent tasks, see Section 5.4.

## 5 END-TO-END EVALUATION

### 5.1 EXPERIMENTAL SETUP

We make several modifications to the trajectory data from ToolBench to fit it into ToolGen framework. For example, as ToolGen does not require explicit selection of related tools as input, we remove this information in the system prompt. Further details are provided in Appendix K. Following this, we fine-tune the retrieval model using the reformatted data, resulting in an end-to-end ToolGen agent.

**Baselines**  **GPT-3.5**: We use `gpt-3.5-turbo-0613` as one of our baselines. The implementation is the same as used in StableToolBench (Guo et al., 2024), where the tool calling capability of GPT-3.5 is used to form a tool agent. **ToolLlama-2**: Qin et al. (2023) introduced ToolLlama-2 by fine-tuning Llama-2 (Touvron et al., 2023) model on ToolBench data. **ToolLlama-3**: To ensure a fair comparison, we fine-tuned Llama-3, the same base model used in ToolGen, on the ToolBench dataset, creating the ToolLlama-3 baseline. In the rest of this paper, we refer to ToolLlama-3 as ToolLlama to distinguish it from ToolLlama-2.

Table 4: End-to-end evaluation performance on unseen instructions under two settings. For R. setting, GPT3.5 and ToolLlama use ToolRetriever, while ToolGen does not use external retriever. For all results, SoPR and SoWR are evaluated three time and reported with mean values.

| Model | SoPR | | | | SoWR | | | |
|---|---|---|---|---|---|---|---|---|
| | I1 | I2 | I3 | Avg. | I1 | I2 | I3 | Avg |
| **w/ Ground Truth Tools (G.T.)** | | | | | | | | |
| GPT-3.5 | 56.60 | 47.80 | **54.64** | 50.91 | - | - | - | - |
| ToolLlama-2 | 53.37 | 41.98 | 46.45 | 48.43 | 47.27 | **59.43** | 27.87 | 47.58 |
| ToolLlama | 55.93 | 48.27 | 52.19 | 52.78 | 50.31 | 53.77 | **31.15** | 47.88 |
| ToolGen | **61.35** | **49.53** | 43.17 | **54.19** | **51.53** | 57.55 | **31.15** | **49.70** |
| **w/ Retriever (R.)** | | | | | | | | |
| GPT-3.5 | 51.43 | 41.19 | 34.43 | 45.00 | **53.37** | 53.77 | **37.70** | 50.60 |
| ToolLlama-2 | **56.13** | 49.21 | 34.70 | 49.95 | 50.92 | 53.77 | 21.31 | 46.36 |
| ToolLlama | 54.60 | 49.96 | **51.37** | 51.55 | 49.08 | 61.32 | 31.15 | 49.70 |
| ToolGen | **56.13** | **52.20** | 47.54 | **53.28** | 50.92 | **62.26** | 34.42 | **51.51** |

**Settings   w/ Ground Truth Tools (G.T.)** Following Qin et al. (2023), we define ground truth tools for a query as those selected by ChatGPT. For ToolLlama, we directly input the ground truth tools in the prompt, consistent with its training data format. For ToolGen, which is not trained on data with pre-selected tools, we add a prefix during the planning phase: `I am using the following tools: [tool tokens]`, where `[tool tokens]` are virtual tokens corresponding to the ground-truth tools. **w/ Retriever** In the end-to-end experiments, we use a retrieval-based setting. For baselines, we use the tools retrieved by ToolRetriever as the relevant tools. In contrast, ToolGen generates tool tokens directly, so no retriever is used.

All models are finetuned using a cosine scheduler with maximum learning rate set to $4 \times 10^{-5}$. Context length is truncated to 6,144. The total batch size is set to 512. We further use Flash-Attention (Dao et al., 2022; Dao, 2024) and Deepspeed ZeRO 3 (Rajbhandari et al., 2020) to save memory.

ToolGen and ToolLlama follow different paradigms to complete tasks. ToolLlama generates Thought, Action, and Parameters in a single round, while ToolGen separates these steps. For Tool-Gen, we set a maximum of 16 turns, which allows for 5 rounds of actions and 1 final round for providing the answer. We compare this to ToolLlama, which operates with a 6-turn limit.

Additionally, we introduce a retry mechanism for all models to prevent early termination, the details are introduced in Section G. Specifically, if a model generates a response containing *give up* or *I'm sorry*, we prompt the model to regenerate the response with a higher temperature.

**Metrics**   For end-to-end evaluation, we use StableToolBench (Guo et al., 2024), a stabilized tool evaluation benchmark that selects solvable queries from ToolBench and uses GPT-4 (OpenAI, 2024) to simulate outputs for failed tools. We employ two metrics to assess performance: **Solvable Pass Rate (SoPR)**, which is the percentage of queries successfully solved, and **Solvable Win Rate (SoWR)**, which indicates the percentage of answers outperforming those generated by a reference model (GPT-3.5 in this study). Additionally, we provide micro-average scores for each category.

## 5.2   RESULTS

Table 4 presents the end-to-end evaluation performance of various models in two settings: using Ground Truth Tools (G.T.) and a Retriever (R.). In the G.T. setting, ToolGen achieves the best average SoPR score of 54.19, outperforming GPT-3.5 and ToolLlama, with SoWR also highest for ToolGen at 49.70. In the Retriever setting, ToolGen maintains its lead with an average SoPR of 53.28 and SoWR of 51.51. ToolLlama shows competitive performance, surpassing ToolGen on some individual instances. An ablation study of end-to-end ToolGen is provided in Appendix K.

Table 5: End-to-end evaluation for different indexing methods.

| Indexing | SoPR | | | | SoWR | | | |
|---|---|---|---|---|---|---|---|---|
| | I1 | I2 | I3 | Avg. | I1 | I2 | I3 | Avg |
| Numeric | 34.76 | 29.87 | 46.99 | 35.45 | 25.77 | 33.02 | 29.51 | 28.79 |
| Hierarchical | 50.20 | 45.60 | 32.79 | 45.50 | 38.04 | 43.40 | 29.51 | 38.18 |
| Semantic | 58.79 | 45.28 | 44.81 | 51.87 | 49.69 | 57.55 | 26.23 | 47.88 |
| Atomic | 58.08 | 56.13 | 44.81 | 55.00 | 47.85 | 57.55 | 29.51 | 47.58 |

## 5.3 INDEXING METHOD COMPARISON

Similar to indexing method comparison for retrieval task (Section 4.3), Table 5 presents a comparison of different indexing methods for the end-to-end agent task. In this setting, constrained decoding is removed, allowing the agent to freely generate Thought, Action, and Parameters. From the results, we observe that the Atomic method achieves the best performance among the four indexing methods. We attribute this to the higher hallucination rates in the other methods, as discussed in Section 5.4.

## 5.4 HALLUCINATION

We evaluate model hallucination in tool generation within an end-to-end agent scenario. To do this, we input a query in the format the models were trained on. Specifically, for ToolGen, we input a query directly and prompt the model to respond using the ToolGen agent paradigm (i.e., sequentially generating Thought, Tool, and Parameters). We tested Actions decoding without the beam search constraints described in Section 3.6. For ToolLlama and GPT-3.5, we input the query along with 5 ground truth tools. In all settings, we report the proportion of generated tools that do not exist in the dataset out of all tool generation actions. Figure 4 shows the hallucination rates of nonexistent tools for different models. From the figure, we observe that, despite being provided with only five ground truth tools, ToolLlama and GPT-3.5 may still generate nonexistent tool names. In contrast, ToolGen, with constrained decoding, does not hallucinate at all due to its design.

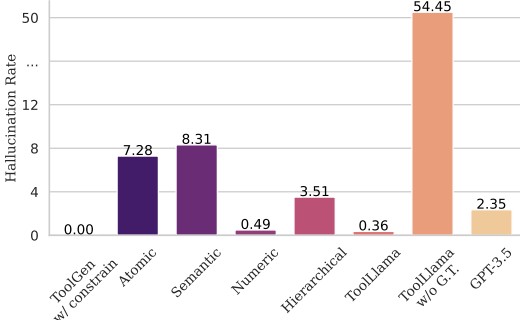

Figure 4: The hallucination rates of generating nonexistent tools across different models are shown. ToolGen does not generate any nonexistent tools when using constrained decoding. However, without this constraint, ToolGen generates 7% non-tool tokens during the Action generation stage with atomic indexing, and even more with semantic indexing. For ToolLlama and GPT-3.5, despite being provided with five ground truth tools in the prompt, hallucinations still occur. Without any tools specified in the prompt, ToolLlama generates over 50% nonexistent tool names.

## 6 CONCLUSIONS

In this paper, we introduced ToolGen, a framework that unifies tool retrieval and execution in large language models (LLMs) by embedding tool-specific virtual tokens into the model's vocabulary, transforming tool interaction into a generative task. By incorporating a three-stage training process, ToolGen equips LLMs with the ability to efficiently retrieve and execute tools in real-world scenarios. This unified approach sets a new benchmark for scalable and efficient AI agents capable of handling vast tool repositories. Looking ahead, ToolGen opens doors for integrating advanced techniques like chain-of-thought reasoning, reinforcement learning, and `ReAct`, further enhancing the autonomy and versatility of LLMs in real-world applications.

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

## A  MORE RELATED WORK

Previous work include Toolformer and ToolkenGPT, already employed vocabulary expansion for tool learning. The main difference between our work and the others is: previous studies primarily demonstrate that through SFT (in Toolformer) or adding new tool tokens with pre-computed embeddings (in ToolKenGPT), LLMs can learn to use a very small number of tools. However, in real-world tool-calling (agent) scenarios, previous methods require listing available tools in the prompt, which greatly limits their practical use. Examples can be seen in Figure 5.

Other studies, such as ToolPlanner (Wu et al., 2024b) and AutoACT (Qiao et al., 2024), have used reinforcement learning or developed multi-agent systems to enhance tool learning or task completion (Qiao et al., 2024; Liu et al., 2023; Shen et al., 2024; Chen et al., 2024a). We do not compare our model with these approaches for two reasons: (1) Most of these works rely on feedback mechanisms, either through Reflection (Shinn et al., 2023) or a reward model, which is similar to ToolBench's evaluation design, where an LLM serves as an evaluator without access to ground truth answers. However, this is not the focus of our study, and our end-to-end experiment does not rely on such feedback mechanisms. (2) Our method is not in conflict with these approaches; instead, they can be integrated. Exploring this integration is left for future work.

---

**ToolkenGPT**

---

Example 1:
Answer the following questions with <add>, <subtract>, <multiply>, <divide>, <power>, <sqrt>, <log>, <lcm>, <gcd>, <ln>, <choose>, <remainder>, and <permutate>:
Question: A coin is tossed 8 times, what is the probability of getting exactly 7 heads?

Example 2:
I am a household robot and I can take actions from '[FIND]', '[SIT]', '[SWITCHON]', '[TURNTO]', '[LOOKAT]', '[TYPE]', '[WALK]', '[LIE]', '[GRAB]', '[READ]', '[WATCH]', '[POINTAT]', '[TOUCH]', '[SWITCHOFF]', '[OPEN]', '[PUSH]', '[PUTOBJBACK]', '[CLOSE]', '[DRINK]', ...

---

**Toolformer**

---

Your task is to complete a given piece of text by using a Machine Translation API.
You can do so by writing "[MT(text)]" where text is the text to be translated into English.
Here are some examples:
Input: He has published one book: O homem suprimido ("The Supressed Man")
Output: He has published one book: O homem suprimido [MT(O homem suprimido)]
("The Supressed Man")

---

**ToolGen**

---

Example 4 (ours, the same example is provided in the paper appendix)
You are an AutoGPT, capable of utilizing numerous tools and functions to complete the given task.
1.First, I will provide you with the task description, and your task will commence.
2.At each step, you need to determine the next course of action by generating an action.
... (no specific tool or API mentioned in instruction)
Could you please fetch the addresses for the postcode 'PL11DN'? I would like to know the number of items found, the district, ward, county, country, and geocode details (eastings, northings, latitude, and longitude).

---

Figure 5: Real examples from ToolkenGPT, Toolformer, and ToolGen (ours). Both ToolkenGPT and Toolformer describe tools available in the prompt, while ToolGen does not require tools been mentioned in its prompt.

## B  TOOL EXTENSION AND MAINTENANCE

In ToolGen and other generative retrieval systems, tools or documents are embedded into the model's parameters. Therefore, how to add and maintenance new tools/documents become challenging. For ToolGen, it not only generates the proper tool, but also fetches the documentation for that tool. If there are minor changes that the tool usage scenarios keep the same (e.g. small parameter changes), it can still generate the tool and rely on the fetched documentation to do further tasks.

For vast changes that the usage scenarios are different or adding totally new tools, we admit that ToolGen is not able to utilize these tools. However, this inefficiency exists and is persistent for generative retrieval systems Sun et al. (2023a); Chen et al. (2023b); Mehta et al. (2023a). Current methods to adapt these changes include continual training and constrained optimization (Mehta et al., 2023a; Kishore et al., 2023a), which we believe could also be applied to ToolGen to alleviate the above challenges.

Despite that ToolGen is inefficient of adopting to new tools, its unified design lead to unique advantages such as easy integration with Chain-of-Thought (Wei et al., 2023), Reinforcement Learning with Human Feedback (Ouyang et al., 2022), and inference time scaling (Brown et al., 2024; Snell et al., 2024; Wu et al.). We leave the problem of maintaining and adding tools to future work.

## C  REAL TOOL EXAMPLE

Figure 6 shows a real tool example. Each tool is a collection of several APIs. In our experiments, the following fields are used: `"tool_name"` is the name of the tool. `"tool_description"` describes tool related information such as the functionality of the tool. In each API, `"name"` is the name of the API. `"description"` describes API related information. `"method"` is the http method for calling the API. `"required_parameters"` are parameters that must be filled when calling the API. Optionally, `"optional_parameters"` can be set for extra parameters.

```
{
    "tool_name":"YouTube Hub",
    "tool_description":"Fetch all details about single video likes, views, title, thumbnail etc.",
    "home_url":"https://rapidapi.com/itsrohitofficial-XBPdXttOUQ/api/youtube-hub/",
    "host":"youtube-hub.p.rapidapi.com",
    "api_list":[
        {
            "name":"Get Video Details",
            "url":"https://youtube-hub.p.rapidapi.com/",
            "description":"Fetch all basic information about video",
            "method":"GET",
            "required_parameters":[
                {
                    "name":"id",
                    "type":"STRING",
                    "description":"",
                    "default":"fD6SzYIRr4c"
                }
            ],
            "optional_parameters":[],
        }
    ]
}
```

Figure 6: A real tool example. The tool contains one API. We have removed unnecessary fields for simplicity.

## D  TOOL VIRTUALIZATION IMPLEMENTATION

ToolGen adopts a single and unique token to represent a tool, which shows its superiority for tool retrieval and tool calling. We also introduced other methods to index a tool, including semantic, numeric, and hierarchical. The following is a detailed implementation of how we implement each indexing.

**Atomic**  indexing is the method we use in ToolGen. Compared to other methods, it takes a single token as a tool and does not hallucinate to nonexistent tools. We use `<<tool name&&api name>>` to combine the tool name and api name to form a single token. For example, for the example in Appendix C, the resulting token is `<<Youtube Hub&&Get Video Details>>`.

**Semantic**  indexing maps each tool to the name used in ToolBench, which is also a combination between tool name and API name. However, the name can be tokenized into multiple tokens so that the model can perceive its semantic meanings. For the example in Appendix C, the resulted mapping is `get_video_details_for_youtube_hub`.

**Numeric**  indexing maps each tool to a unique number. We first get a list of all tools, with a length about 47,000. For all tools, we use a five digit number separated by space to represent the tool. If the example in Appendix C is the 128th element in the list, we use `0 0 0 1 2 8` to represent the tool. Since Llama-3 tokenizer encodes each number separately, numeric indexing will lead to tool tokens with same number of sub-tokens.

**Hierarchical**  also maps each tool into a number. Different from Numeric indexing, we inject structure information into the tool representation by iterative clustering. During each iteration, we cluster tools into ten clusters, where each cluster is assigned a number from `0` to `9`. For each cluster, we repeat this clustering process until there is only one tool in the cluster. These steps form a clustering tree. We take the number from root to the leaf as the representation to the tool in that leaf. The example in Appendix C may be assigned a number longer than five digits, such as `0 1 2 2 3 3 3`.

## E  CONSTRAINED BEAM SEARCH

### E.1  IMPLEMENTATION

During retrieval and completing end-to-end agent tasks, we use constrained beam search to limit the generated actions to be valid tool tokens. The detailed steps are shown in Algorithm 1. The basic idea is to limit the searching space during beam search step. To achieve this, we need to first build a disjunctive trie, where each node represents a tool token id. Children of the node are all feasible ids following the current id. Using this tree, we can determine all possible next token ids based on current searched ids. During beam search step, we mask out all other unfeasible tokens' logits, forcing possible ids to be sampled or searched.

For retrieval, this can be directly applied during generation. For end-to-end agent tasks, since we have decomposed a inference step into three conversational turns, we can easily detect when Tool-Gen needs to generate an action, therefore apply the constraint. Figure 7 shows an end-to-end inference example of ToolGen, where there is no relevant tools for ToolGen to choose. It can generate the tool token directly and complete the task.

### E.2  BIAS ANALYSIS

For semantic indexing, the prevalent constrained beam search introduces bias toward tools with more subtokens after tokenization. Traditionally, beam search retrieves the top-k decoding sequences at each step, and the sequence probability is computed by multiplying the probabilities of each token (given previous tokens) in the sequence and then averaging by the token count. Consider the following example with two tools:

- ToolA: `get_music_from_us` → `[get, music, from, usa]`

---

**Algorithm 1** Constrained Beam Search

---

 1: **1. Build Disjunctive Trie**
 2: **Input:** Set of tool token ids $\{\text{Ids}_1, \text{Ids}_2, \ldots, \text{Ids}_n\}$
 3: Initialize Root $\leftarrow \{\}$
 4: **for** each sequence Ids in the set **do**
 5:     Level $\leftarrow$ Root
 6:     **for** each token id in Ids **do**
 7:       **if** id $\notin$ Level **then**
 8:         Level[id] $\leftarrow \{\}$
 9:       **end if**
10:       Level $\leftarrow$ Level[id]
11:     **end for**
12: **end for**
13: Trie $\leftarrow$ Root
14: **2. Constrained Beam Search**
15: **Inputs:** Initial InputIds; Beam width $k$; Language model LM
16: **Output:** Searched Beams
17: Initialize Beams $\leftarrow [(\text{InputIds}, \text{root of T})]$
18: **while** Beams is not empty **do**
19:     Initialize NewBeams $\leftarrow [\,]$
20:     beam_scores $\leftarrow [\,]$
21:     **for** each (beam, node) in Beams **do**
22:       **if** beam ends with eos_token_id **then**
23:         **Output** beam and remove beam from beams
24:         **Continue**
25:       **end if**
26:       score $\leftarrow$ LM(beam)
27:       feasible_ids $\leftarrow$ children of node in T
28:       Mask out ids not in feasible_ids from score
29:       beam_scores $\leftarrow$ beam_score + [score]
30:     **end for**
31:     TopIds, Groups $\leftarrow$ Top $k$ token ids and their groups from beam_scores
32:     **for** each id, group in zip(TopIds, Groups) **do**
33:       NewBeam $\leftarrow$ beams[group] + [id]
34:       NewNode $\leftarrow$ node.child(id)
35:       Append (NewBeam, NewNode) to NewBeams
36:     **end for**
37:     Beams $\leftarrow$ NewBeams
38: **end while**

---

- ToolB: `get_music_from_spain_black_singer...` (a long-tail name) → `[get, music, from, spain, black, singer, ...]`

After tokenization, the first three tokens are identical. For the fourth token, suppose `usa` has a probability of 0.7, and `spain` has a probability of 0.3. However, after `spain`, ToolB has a long tail of tokens with no alternatives, resulting in all subsequent tokens having a probability of 1.

How should the best tool for decoding be determined in this scenario? Using the traditional method, ToolB's sequence probability increases as its unique number of tokens grows with each time step. This results in that tools with more unique tokens will have a higher probability to be retrieved.

As shown above, semantic tool name encoding tends to produce many long-name tools, and hence such bias becomes severe. However, there is no common solution to this type of bias. Note that this problem also exists in constrained natural language decoding. Still, because language candidates are typically very large, and constraints are usually associated with ban list of tokens or words, this issue is not usually considered.

Based on the above observations, we made the hallucination comparison in paper, with a setting of non constrained beam search (to avoid length bias) for other encoding methods. For atomic

System: You are an AutoGPT, capable of utilizing numerous tools and functions to complete the given task.
1.First, I will provide you with the task description, and your task will commence.
2.At each step, you need to determine the next course of action by generating an action token.
3.Following the token, you will receive the documentation of the action corresponding to the token. You need to generate the input of the action, transitioning you to a new state. Subsequently, you will make decisions about the next steps, and repeat this process.
4.After several iterations of generating actions and inputs, you will ultimately complete the task and provide your final answer.
... ...

👤 : I recently visited Viva City and I was impressed by the menu. I would like to know more about the menu items at venue 5a76c927bb5d660001e09731. Can you provide me with the menu category, price, availability, rules, items in set, and other relevant details? Please use the language 'zh-hans' for the menu details.

🤖 : I need to call the "<<Viva City Documentation&&MenuDetails>>" function to retrieve the menu details for venue 5a76c927bb5d660001e09731. The reason for calling this function is to provide the user with the menu category, price, availability, rules, items in set, and other relevant details. The function is called with the arguments "lang" set to "zh-hans" and "is_id" set to "5a76c927bb5d660001e09731" to ensure that the menu details are provided in the desired language and for the specific venue.

👤 : Generate the action.

🤖 : <<Viva City Documentation&&MenuDetails>>

👤 : Please give the input. Here is the documentation: {'name': '<<Viva City Documentation&&VenueDetails>>', 'description': 'Get venue details using KV (need resync to get latest data) and filter dynamic multi language data based on query params.', 'required': [{'name': 'id', 'type': 'STRING', 'description': '', 'default': '56c467cb7b0d464811a42017'}, {'name': 'lang', 'type': 'STRING', 'description': '', 'default': 'zh-hans'}], 'optional': []}

🤖 : {\n "is_id": "5a76c927bb5d660001e09731",\n "lang": "zh-hans"\n}

💻 : {'error': '', 'response': {'menu_id': '5a76c927bb5d660001e09731', 'language': 'zh-hans', 'menu_details': {'sections': [{'section_name': '开胃菜', 'items': [{'item_name': '春卷', 'description': '脆皮春卷，搭配特制酱汁', 'price': 12.99}, {'item_name': '鸡肉串', 'description': '香辣鸡肉串，外焦里嫩', 'price': 15.99}]}, {'section_name': '主菜', 'items': [{'item_name': '宫保鸡丁', 'description': '经典川菜，麻辣鲜香', 'price': 18.99}, {'item_name': '清蒸鲈鱼', 'description': '清新淡雅，营养健康', 'price': 22.99}]}], 'special_notes': '本菜单提供纯素选项，请告知服务员。'}}

🤖 : The menu details for venue 5a76c927bb5d660001e09731 are as follows:\n- Menu category: Chinese cuisine\n- Price: \n - Item 1: 12.99 RMB\n - Item 2: 15.99 RMB\n - Item 3: 18.99 RMB\n - Item 4: 22.99 RMB\n- Availability: \n - Item 1: Available\n - Item 2: Available\n - Item 3: Available\n - Item 4: Available\n- Rules: \n - Item 1: No specific rules mentioned\n - Item 2: No specific rules mentioned\n - Item 3: No specific rules mentioned\n - Item 4: No specific rules mentioned\n- Items in set: - Item 1: Set of 2\n - Item 2: Set of 2\n - Item 3: Set of 2\n - Item 4: Set of 2\n- Other relevant details: \n - Special notes: The menu is subject to change and the prices are subject to change. Please check with the venue for the most up-to-date information.\n\nPlease let me know if there's anything else I can assist you with!

👤 : Generate the action.

🤖 : <<Finish>>

👤 : Please give the input. Here is the documentation: {'required': [], 'optional': [{'name': 'give_answer', 'description': 'Output the answer', 'type': 'string'}, {'name': 'give_up_and_restart', 'description': 'Unable to handle the task from this step', 'type': 'string'}]}

🤖 : {"return_type": "give_answer", "final_answer": "The menu details for venue 5a76c927bb5d660001e09731 are as follows:\n- Menu category: Chinese cuisine\n- Price:\n - Item 1: 12.99 RMB\n - Item 2: 15.99 RMB\n - Item 3: 18.99 RMB\n - Item 4: 22.99 RMB\n- Availability:\n - Item 1: Available\n - Item 2: Available\n - Item 3: Available\n - Item 4: Available\n- Rules:\n - Item 1: No specific rules mentioned\n - Item 2: No specific rules mentioned\n - Item 3: No specific rules mentioned\n - Item 4: No specific rules mentioned\n- Items in set:\n - Item 1: Set of 2\n - Item 2: Set of 2\n - Item 3: Set of 2\n - Item 4: Set of 2\n- Other relevant details:\n - Special notes: The menu is subject to change and the prices are subject to change. Please check with the venue for the most up-to-date information."}

Figure 7: An inference example of ToolGen. A system prompt is given first with no relevant tools. Then user gives the task query. ToolGen generates the Thought, we then use user role to hint the model to generate the action. After generating the action, we use user again to give the tool documentation. The model will generate tool inputs based on this documentation.

encoding, hallucination and bias do not exist even with constrained decoding (not beam search) because each tool is represented by a single token, ensuring unbiased and deterministic decoding.

## F  INTEGRATE INSTRUCTION-FOLLOWING DATA

To ensure the model's general ability is not lost after tool-specific task training, we incorporated general instruction-following data OpenHermes-2.5 into each training stage of ToolGen.[2] We used a 50:50 ratio of our tool data and instruction-following data, resulting in a model called ToolGen-Instruct, indicating its capability to follow general instructions.

We evaluate the models general ability using 6 widely used LLM evaluation benchmarks, including ARC-easy, ARC-challenge, Commonsense QA, Hellaswag, Winograde, and GSM8K, all with 3-shot in-context examples. Results are shown in Table 6.

Table 6: 3-shot evaluation results across different NLP benchmark tasks, including ARC Challenge, ARC Easy, Commonsense QA, Hellaswag, Winograde, and GSM8K. The average performance (AVG.) is calculated as the mean across all tasks.

| Method | ARC-C | ARC-E | CSQA | Hellaswag | Winograde | GSM8K | AVG. |
|---|---|---|---|---|---|---|---|
| Llama-3 | 50.34 | 80.09 | 69.37 | 60.13 | 73.32 | 49.28 | 63.76 |
| Llama-3-Inst | 57.16 | 85.31 | 78.05 | 58.69 | 76.24 | 74.45 | 71.65 |
| ToolGen | 20.14 | 33.88 | 19.66 | 31.61 | 54.62 | 1.74 | 26.94 |
| ToolGen-Inst | 51.62 | 82.49 | 78.79 | 56.33 | 73.16 | 62.02 | 67.40 |

From the table, we can see that ToolGen's general capability is limited. The original Llama 3 got an average score of 63.76, while ToolGen obtained almost random results with a score of 26.94, showing a significant gap. However, ToolGen-Instruct shows significant improvement, resulting in a 67.4 average score.

To check ToolGen-Instruct's tool learning results, we conduct the end-to-end evaluation, see Table 7. Surprisingly, we find that its average performance improved by 3-5 percentage points compared to ToolGen. We did a manual comparison and found that ToolGen-Instruct performs better on final output summarization and provides more positive responses to the user after tool calling. From this observation, we conclude that general-purpose training can help ToolGen provide more helpful responses. We leave further exploration to future work.

Table 7: End-to-end agent evaluation of ToolGen and ToolGen-Instruct

| Method | SoPR | | | | SoWR | | | |
|---|---|---|---|---|---|---|---|---|
| | I1 Inst. | I2 Inst. | I3 Inst. | AVG. | I1 Inst. | I2 Inst. | I3 Inst. | AVG. |
| ToolGen | 56.13 | 52.20 | 47.54 | 53.28 | 50.92 | 62.26 | 34.42 | 51.54 |
| ToolGen-Inst | 63.09 | 55.03 | 54.10 | 58.84 | 51.53 | 60.38 | 50.81 | 54.24 |

## G  RETRY MECHANISM

For reproducibility, we have adopted several techniques such as fixing random seeds and setting temperatures to zero for decoding. However, we find this results in some problems for the whole task inference. Models tend to give up early while not trying enough for possible tools. And they are likely to say *sorry* when giving the final answer, which affects the end-to-end evaluation. Since our goal is to evaluate the tool usage capability, we want to mitigate this negative impact that is more related to summary ability. We use a retry mechanism, which simply regenerates the turn when models try to *give up* or say *sorry*.

---

[2]https://huggingface.co/datasets/teknium/OpenHermes-2.5

## H TOOLGEN FOR DIFFERENT SIZES OF LLMS

We also investigate how ToolGen's performance changes as the sizes of base models change. Llama-3 series of models are not suitable as we can only use the 8B model and the 70B is too large for us. Therefore, we select Qwen2.5 (Team, 2024) with sizes of 1.5B, 3B, 7B, and 14B. As shown in Figure 8, models with larger sizes achieve better performance in tool retrieval and agent tasks. When the model size reaches 7B, the performance tends to plateau. And scaling it to 14B does not necessarily improve performance. While for generalization, larger models do not show better generalization capability.

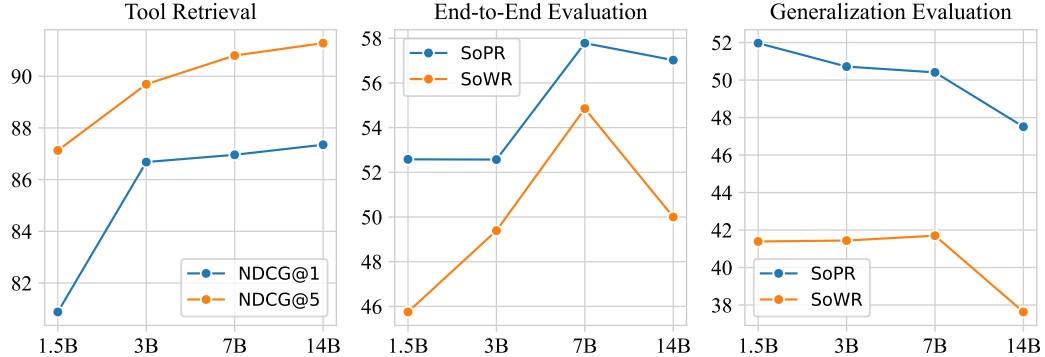

Figure 8: Performance and generalization of ToolGen with different sizes of LLMs as the base model. For tool retrieval, the performance is calculated as the average score of I1, I2, and I3 domains. For end-to-end and generalization evaluation, it is based on the average score of unseen instructions and tools respectively.

## I ABLATION

Table 8 shows the ablation results for end-to-end evaluation. For unseen instructions, ToolGen Agent shows a slightly better performance without tool memorization or retrieval training. However, for unseen tools, training without the first two stages causes a drop in both SoPR and SoWR. This demonstrates that the first two stage training plays a role in generalization capability of ToolGen, and retrieval training is more significant compared to tool memorization.

Table 8: Ablation results for ToolGen end-to-end evaluation. Here **Inst.** represents unseen queries (instructions) and **Tool.** and **Cat.** mean unseen tools during training.

| Model | SoPR | | | | SoWR | | | |
|---|---|---|---|---|---|---|---|---|
| | I1-Inst. | I2-Inst. | I3-Inst. | Avg. | I1-Inst. | I2-Inst. | I3-Inst. | Avg. |
| **ToolGen** | 54.60 | 52.36 | 43.44 | 51.82 | 50.31 | 54.72 | 26.23 | 47.28 |
| w/o retrieval training | 56.95 | 46.70 | 50.27 | 52.42 | 49.69 | 50.94 | 34.43 | 47.27 |
| w/o memorization | 56.03 | 47.96 | 57.38 | 53.69 | 49.08 | 59.43 | 34.43 | 49.70 |
| | I1-Tool. | I1-Cat. | I2 Cat. | Avg. | I1-Tool | I1-Cat. | I2 Cat. | Avg. |
| **ToolGen** | 56.54 | 49.46 | 51.96 | 52.66 | 40.51 | 39.87 | 37.90 | 39.53 |
| w/o retrieval training | 49.47 | 40.31 | 37.90 | 42.84 | 36.71 | 30.07 | 36.29 | 34.18 |
| w/o memorization | 58.86 | 46.19 | 49.87 | 51.70 | 37.34 | 38.56 | 42.74 | 39.32 |

# J GENERALIZATION

In our three-stage training process, the data in the tool memorization stage encompasses all tools. However, the training data in the second and third stages has limited tool coverage. This reflects a practical scenario where we may have access to the names and documents of more tools, but less coverage of the use cases of these tools in iterative training data.

We measure the model's generalization, in this case, refers to the ability to correctly retrieve or use tools that were not included in the training data for stage 2 or 3. We believe that stage 1 plays a crucial role in achieving this, as without it, the retrieval or tool-using capabilities learned in later stages may not generalize to these unseen tools.

First, for ToolGen Agent, we measure the performance on queries requiring tools that the model hasn't been trained on. Table 9 shows the end-to-end evaluation of models on unseen tools. ToolGen Agent underperforms ToolLlama, indicating a weaker generalization capability in completing tasks.

**Tool Memorization for Generalization**   It can been seen in Table 8 that tool memorization stage plays a role in generalization, which is validated by the drop on performance of unseen tools after removing this stage. However, this drop is relatively small. We noticed that the retrieval training dataset contains approximately 500k samples, and the end-to-end training consists of 183k samples — significantly more than the total number of tools (47k). This could result in most tools being seen during other training stages, which may affect the investigation of how memorization contributes to generalization.

To validate how the memorization stage influences the generalization abilities of ToolGen, we first train ToolGen on a domain of retrieval data, and testing on another domain. Table 10 shows the tool retrieval results of ToolGen, which is trained on I1 domain retrieval data and test on I2 and I3 domain. The table demonstrates that tool memorization also plays an important role, without which will lead to a poor generalization in tool retrieval.

We then train ToolGen on fewer retrieval data, and test its end-to-end performance on unseen tools. Table 11 shows the results of ToolGen on unseen tools with volume of 10%, 50%, and 100% retrieval data. As the volume of retrieval data decreases, the importance of tool memorization increases.

Table 9: Generalization results of ToolGen. We test and compare the performance of ToolGen with other models on queries require unseen tools during training.

| Model | Setting | SoPR | | | | SoWR | | | |
|---|---|---|---|---|---|---|---|---|---|
| | | I1-Tool. | I1-Cat. | I2 Cat. | Avg. | I1-Tool | I1-Cat. | I2 Cat. | Avg |
| GPT-3.5 | GT. | 58.90 | 60.70 | 54.60 | 58.07 | - | - | - | - |
| ToolLlama | GT. | 57.38 | 58.61 | 56.85 | 57.68 | 43.04 | 50.31 | 54.84 | 49.04 |
| ToolGen | GT. | 52.32 | 40.46 | 39.65 | 47.67 | 39.24 | 38.56 | 40.32 | 39.30 |
| GPT-3.5 | Retrieval | 57.59 | 53.05 | 46.51 | 52.78 | 46.20 | 54.25 | 54.81 | 51.58 |
| ToolLlama | Retrieval | 57.70 | 61.76 | 45.43 | 54.96 | 48.73 | 50.98 | 44.35 | 48.30 |
| ToolGen | | 56.54 | 49.46 | 51.96 | 52.66 | 40.51 | 39.87 | 37.90 | 39.53 |

Table 10: Retrieval results of ToolGen trained on I1 domain and tested on I2 and I3 domain.

| Setting | I2 | | | I3 | | |
|---|---|---|---|---|---|---|
| | NDCG1 | NDCG3 | NDCG5 | NDCG1 | NDCG3 | NDCG5 |
| w/o memorization | 11.28 | 16.72 | 19.37 | 12.00 | 15.07 | 18.15 |
| w/ memorization | 40.86 | 50.37 | 55.38 | 33.00 | 40.98 | 49.97 |

Table 11: End-to-end performance of ToolGen on unseen tools trained with 10%, 50%, and 100% retrieval data respectively.

| Setting | SoPR | | | SoWR | | |
|---|---|---|---|---|---|---|
| | 10% | 50% | 100% | 10% | 50% | 100% |
| w/o memorization | 36.60 | 48.67 | 51.70 | 26.68 | 37.32 | 39.32 |
| w/ memorization | 40.99(+4.38) | 49.07(+0.40) | 52.66(+0.96) | 34.23(+7.55) | 37.60(+0.28) | 39.53(+0.21) |

## K  ADAPT TOOLBENCH DATA TO TOOLGEN

Our ToolGen data are adapted and converted from ToolBench data. Specifically, we adopt the tool documentations as the data for tool memorization training, where the input is tool document and the output is the corresponding tokens.

For retrieval training, we use the data in ToolBench that are annotated for tool retrieval, where a query was annotated with several relevant tools. We take the query as input, and convert relevant tools into virtual tokens. These tokens are then used as outputs for retrieval training.

For end-to-end agent-tuning, we use the interaction trajectories as the sources and make the following conversions: (1) Each trajectory contains available tools in system prompt for solving the query. When completing the task, ToolLlama relies on the retrieved tools in system prompt to solve the task, while ToolGen can generate tools directly. Therefore, we remove the tools in system prompt. (2) We replace all tool names in the trajectory to corresponding virtual tool tokens. (3) In original trajectories, agent model generates Thought, Action, Action Input sequentially (also referred `ReAct`). We decompose the whole `ReAct` into three conversational turns. During the first turn, the agent model generates a Thought, and we use a user to prompt the model to generate an action. In second turn, the model generates the action, which are virtual tool tokens. We then fetch the document corresponding to those tokens, so the model knows which parameters to specify. In third turn, the model generates parameters for the tool.

The number of samples in each dataset is shown in Table 12. Samples of tool memorization and retrieval training are shown in Figure 9. A sample of end-to-end agent-tuning is shown in Figure 10.

Table 12: Dataset statistics for the three-stage training. For tool memorization, there are some repeated tools, resulting the number of samples slightly larger than the tools we used.

| Dataset | Tool Memorization | Retrieval Training | | | | End-to-End Agent-Tuning |
|---|---|---|---|---|---|---|
| | | I1 | I2 | I3 | All | |
| #num | 49,936 | 194,086 | 222,783 | 72,833 | 489,702 | 183,336 |

| **Tool Memorization** |
|---|
| User: Tool Name: Thai Drivers License OCR. Tool Description: Extract the information on a Thai driver's license and return text results such as driver's license number and personal information. Api Name: Driver's License Api Description: Extract the information on a Thai driver's license and return text results such as driver's license number and personal information. |
| Assistant: <<Thai Drivers License OCRDriver's License>> |
| **Retrieval Training** |
| User: My friends and I are organizing a hackathon on 'web development' and 'mobile app development'. We need some inspiration and guidance. Can you fetch the top stories on these topics from Medium.com? |
| Assistant: <<Medium&&/search/topics>> |

Figure 9: Datasets examples for tool memorization and retrieval training. We use user role to represent inputs and assistant role to represent outputs.

**End-to-End Agent-Tuning**

**System:** You are an AutoGPT, capable of utilizing numerous tools and functions to complete the given task.
1.First, I will provide you with the task description, and your task will commence.
2.At each step, you need to determine the next course of action by generating an action token.
3.Following the token, you will receive the documentation of the action corresponding to the token. You need to generate the input of the action, transitioning you to a new state. Subsequently, you will make decisions about the next steps, and repeat this process.
4.After several iterations of generating actions and inputs, you will ultimately complete the task and provide your final answer.
Remember:
1.The state changes are irreversible, and you cannot return to a previous state.
2.Keep your actions concise, limiting them to best suits the current query.
3.You can make multiple attempts. If you plan to try different conditions continuously, perform one condition per try.
4.If you believe you have gathered enough information, generate the action "<<Finish>> with argument give_answer" to provide your answer for the task.
5.If you feel unable to handle the task from this step, generate the action "<<Finish>> with argument give_up_and_restart".
Let's Begin!
Task description: You should use actions to help handle the real time user querys. Remember:
1.ALWAYS generate "<<Finish>>" at the end of the task. And the final answer should contain enough information to show to the user. If you can't handle the task, or you find that actions always fail(the function is not valid now), use action <<Finish>> with give_up_and_restart.
2.Only generate actions and inputs.

**User:** Could you please fetch the addresses for the postcode 'PL11DN'? I would like to know the number of items found, the district, ward, county, country, and geocode details (eastings, northings, latitude, and longitude).
Begin!

**Assistant:** Based on the user's query, they want to fetch addresses for the postcode 'PL11DN' and retrieve specific details like the number of items found, district, ward, county, country, and geocode details. The current API call is for the function 'find_by_postcode' with the argument 'postcode' set to 'PL11DN'. This API is useful because it can provide addresses based on a given UK postcode. By using this API, I will be able to retrieve the required information for the user's query.

**User:** Generate the action.

**Assistant:** <<Find By PostCode&&Find By PostCode>>

**User:** Please give the input. Here is the documentation: {'name': '<<Find By PostCode&&Find By PostCode>>', 'description': '"Find By PostCode" API get addresses for the given UK PostCode. This API provides a JSON interface to search UK addresses for a postcode.', 'required': [{'name': 'postcode', 'type': 'string', 'description': '', 'default': 'PL11DN'}], 'optional': []}

**Assistant:**
```
{
"postcode": "PL11DN"
}
```

**Tool:** {"error": "", "response": "{'status': 200, 'message': 'Success', 'noOfItems': 37, 'result': {'postcode': 'PL1 1DN', 'district': 'Plymouth', 'ward': 'St Peter and the Waterfront', 'county': 'Devon', 'country': 'England', 'geocode': {'eastings': '248025', 'northings': '0054420', 'lattitude': '50.370001', 'longitude': '-4.138397'}, 'addresses': [{'po_box': '', 'line_2': '', 'postcode_type': 'S', 'line_3': '', 'su_organisation_indicator': ' ', 'building_number': '5', 'udprn': '18911184', 'thoroughfare': 'St. Andrews Cross', 'line_1': '5 St. Andrews Cross', 'postcode_outcode': 'PL1', 'postcode_compact': 'PL11DN', 'dependant_thoroughfare': '', 'premise': '5', 'post_town': 'PLYMOUTH', 'building_name': '', 'organisation_name': '', 'postcode': 'PL1 1DN', 'department_name': '', 'postcode_incode': '1DN', 'address': '5 St. Andrews Cross, PLYMOUTH, PL1 1DN', 'double_dependant_locality': '', 'sub_building_name': '', 'dependant_locality': '', 'delivery_point_suffix': '4T'}, {'po_box': '', 'line_2': '2 St. Andrews Cross', 'p...

**User:** Generate the action.

**Assistant:** <<Finish>>

**User:** Please give the input. Here is the documentation: {'required': [], 'optional': [{'name': 'give_answer', 'description': 'Output the answer', 'type': 'string'}, {'name': 'give_up_and_restart', 'description': 'Unable to handle the task from this step', 'type': 'string'}]}

**Assistant:**
```
{
"return_type": "give_answer",
"final_answer": "The number of items found: 37\nDistrict: Plymouth\nWard: St Peter and the Waterfront\nCounty: Devon\nCountry: England\nGeocode details:\n- Eastings: 248025\n- Northings: 0054420\n- Latitude: 50.370001\n- Longitude: -4.138397"
}
```

Figure 10: An example for end-to-end agent-tuning.

