# OpenReview forum: "ToolGen: Unified Tool Retrieval and Calling via Generation"
_ICLR.cc/2025/Conference — ICLR 2025 Poster_

### Official Review · Reviewer_A87g · 2024-10-31

**Soundness:** 3
**Presentation:** 3
**Contribution:** 4
**Rating:** 8
**Confidence:** 4

**Summary:**

This paper proposes ToolGen, a generative tool/function calling framework. The concrete methods are: (1) virtualizing tools by virtual tokens; (2) memorizing tools with training data of (tool docs, tool virtual tokens); (3) learning tool retrieval with training data of (user queries, tool virtual tokens); (4) and finally, finetuning tool agent with tool calling trajectories. Experimental results with over 47,000 tools show that ToolGen not only achieves superior results in both tool retrieval and autonomous task completion but also sets the stage for a new era of AI agents that can adapt to tools across diverse domains.

**Strengths:**

Overall, I like this paper very much.

1. ToolGen is now paradigm for fundamentally transforming tool retrieval into a generative process, which is a very important topic for function calling or LLM-based agents.
2. The methods are sound and resonable; the paper presentation is clear.
3. The experimental results show that the methods are effective compared with several strong baselines.

**Weaknesses:**

1. Traditional retrieval and generation methods for function calling can handle dynamic tools. If the tool set is changing, could ToolGen be used (without retraining)?

2. A few reltated works are not mentioned or compared. For example, TPTU and TPTU-V2 [1,2,3,] used demo retriever and fintuner besides the tool retrieval, which may be more powerful than the traditional retrieval and generation methods. They can be a strong baseline for comparision.

3. It seems that ToolGen uses a more complex process (i.e., (1) virtualizing tools by virtual tokens; (2) memorizing tools with training data of (tool docs, tool virtual tokens); (3) learning tool retrieval with training data of (user queries, tool virtual tokens); (4) and finally, finetuning tool agent with tool calling trajectories) to prepare for tool calling. Therefore, the suprior of ToolGen may come from more training of LLMs? Could the authors give more explination on this?


[1] TPTU: Task Planning and Tool Usage of Large Language Model-based AI Agents. FMDM Workshop at NeurIPS 2023.
[2] TPTU-v2: Boosting Task Planning and Tool Usage of Large Language Model-based Agents in Real-world Systems. LLMAgents Workshop at ICLR 2024.
[3] TPTU-v2: Boosting Task Planning and Tool Usage of Large Language Model-based Agents in Real-world Industry Systems. EMNLP 2024 Industry Track.

**Questions:**

see the above.

---

> ### Author Response · Authors · 2024-11-19
>
> We sincerely thank you for the positive and encouraging feedback on our work. We appreciate your acknowledgment of the strengths of our paper. We also value your recognition of the importance of transforming tool retrieval into a generative process for advancing LLM-based agents. For your comments and questions, we address as follows:
>
> > Tool set changes
>
> ToolGen is trained to retrieve tools based on their descriptions (i.e. What does the tool do) and user queries. During inference, it first generates the tool to be used, then fetches the document for the tool. Therefore, as long as tool descriptions remain unchanged, ToolGen is still able to generate proper tools with some minor changes (e.g. Tool parameters). However, in situations where the tool changes vastly so that it can not be used in previous scenarios or adding totally new tools, ToolGen does not know how to use them. This problem exists in our work and it is also persistent in other generative retrieval works [1,2,3].
>
> Some popular solutions include continual training [2,3] and constrained optimization [4]. There could also be some other flexible methods. For example, we can reserve some slots in prompts for updated tools. When we want to add maintain a tool or add new tools, we can simply put them in prompts. We could also utilize knowledge editing to maintain an existing tool. Although we take it as a future work, we will add the discussion in the revised version of our paper.
>
> > Unmentioned related works
>
> Thank you for pointing out these works we neglected. After carefully checking the details of TPTU and TPTU-V2 [5,6,7], we find that the settings between these works and ours are different, which makes it difficult to reimplement and compare the results. We are happy to add the following discussion in related work to make it more complete.
> "TPTU [5] proposes a structured framework for LLM agents and evaluates their task planning and tool usage abilities. Furthermore, TPTU-v2 [6,7] builds an LLM Finetuner to enhance agent performance with curated datasets and a demo selector to select relevant demonstrations. They set a flexible and superior paradigm compared to traditional retrieval-based paradigm."
>
> > More explanation on ToolGen's superior performance
> 1. From the perspective of tool retrieval, previous methods generally use a relatively weak model (BM25 or BERT), which have limited parameters compared to ToolGen. Those models generally compress user queries and tools into vectors and select tools based on vector similarity. Such methods may not fully capture the user's intents, details in queries, and functionality in tools, while these may be important to select a proper tool. For ToolGen, we use Llama-3 as the base model, which has a better capability compared to BERT. We make it to memorize all tools into its vast parameters. During inference, all the user query tokens are kept for generating the tool without compression. Therefore, it achieves a far better tool retrieval performance.
> 2. For agent tasks, first of all, the selected tools of previous methods are not as good as ToolGen's generated tool. ToolGen is not only trained on agent tuning datasets, but are also enhanced with previous tool memorization and retrieval training. Therefore, its tool usage capability is also better, leading to better end-to-end performance.
>
> Reference:
>
> [1] Sun, Weiwei, et al. "Learning to tokenize for generative retrieval." Advances in Neural Information Processing Systems 36 (2024).
>
> [2] Chen, Jiangui, et al. "Continual learning for generative retrieval over dynamic corpora." Proceedings of the 32nd ACM International Conference on Information and Knowledge Management. 2023.
>
> [3] Mehta, Sanket Vaibhav, et al. "DSI++: Updating Transformer Memory with New Documents." Proceedings of the 2023 Conference on Empirical Methods in Natural Language Processing. 2023.
>
> [4] Kishore, Varsha, et al. "Incdsi: incrementally updatable document retrieval." International Conference on Machine Learning. PMLR, 2023.
>
> [5] Ruan, Jingqing, et al. "Tptu: Task planning and tool usage of large language model-based ai agents." NeurIPS 2023 Foundation Models for Decision Making Workshop. 2023.
>
> [6] Kong, Yilun, et al. "TPTU-v2: Boosting Task Planning and Tool Usage of Large Language Model-based Agents in Real-world Industry Systems." Proceedings of the 2024 Conference on Empirical Methods in Natural Language Processing: Industry Track. 2024.
>
> [7] Kong, Yilun, et al. "TPTU-v2: Boosting Task Planning and Tool Usage of Large Language Model-based Agents in Real-world Systems." ICLR 2024 Workshop on Large Language Model (LLM) Agents.

---

> > ### Author Response · Authors · 2024-11-23
> >
> > We sincerely thank you again for your recognition of our work. As we mentioned in the previous response, we have revised the paper and added it in related work (L139).

---

> > ### Comment · Reviewer_A87g · 2024-11-26
> > **I support the acceptance of this paper, and I will keep my score**
> >
> > As I mentioned in the reivew, this paper present a novel perspective for tool calling - the generative tool calling, and the authors prove this is possiable with good results. I think this new method (in fact a new direction!) is imprtant than other details. So support the acceptance of this paper. Since my score is high, I will keep my score.
> >
> > ps: I read the reviews from other reviewers, I agree with that this paper has some aspects to improve, but this did not change my score.

---

### Official Review · Reviewer_SiCu · 2024-11-02

**Soundness:** 1
**Presentation:** 3
**Contribution:** 2
**Rating:** 5
**Confidence:** 3

**Summary:**

The paper introduces ToolGen, a novel framework designed to address the limitations of LLMs in interacting with an extensive array of external tools. This framework shifts existing paradigms by embedding knowledge about various tools directly into the LLM’s vocabulary, using unique tokens to represent each tool. This allows the model to generate tool calls and arguments as part of the language generation process, eliminating the need for separate tool retrieval systems and seamlessly integrating tool usage within LLM capabilities. ToolGen incorporates a three-stage training process to enable the LLM to learn how to use tokenized tools.

The main contribution of this paper is introducing the ToolGen framework including training and inferencing, and proving that ToolGen have comparable result with current tool retrieval systems.

**Strengths:**

**Originality:**
The paper introduces a novel approach to tool retrieval by representing each tool as a unique virtual token directly integrated into the LLM’s vocabulary. This method eliminates the need for auxiliary retrievers, making the retrieval process more seamless and efficient. The concept of transforming tool retrieval into a generative task is novel and presents a possible solution to the scalability challenges faced by some existing methods.

**Quality:**
The authors have conducted a comprehensive set of experiments to validate their claims, showing thoroughness in their evaluation. These experiments include various comparisons and detailed ablation studies to assess the impact of different components of their approach. The robustness of their methodology enhances the credibility of their results and conclusions.

**Clarity:**
The paper is well-structured and written in a clear and concise manner. It systematically outlines the problem, the proposed solution, and the experimental validations. Each section transitions smoothly into the next, making it easy for the reader to follow the logic and understand the contributions of the work.

**Significance:**
The research addresses a critical issue in the field of LLM agents: managing and retrieving tools from a large set of tools efficiently. Given the increasing complexity and number of tools available, the proposed method provides a solution. This work has implications for the development of more autonomous and efficient AI systems, potentially benefiting some applications where tool interaction is essential.

**Weaknesses:**

**Substantive Assessment of Weaknesses:**

**Cost and Efficiency Claims:**
The key claim of "significantly less cost and higher efficiency" does not hold up under scrutiny. The authors have not substantively demonstrated that their framework is less costly or more efficient than existing methodologies. The ToolGen framework necessitates a three-stage training process, which does not inherently suggest reduced costs. Furthermore, the paper lack data or experiments to substantiate the claim regarding efficiency improvements over other approaches.

**Role of the Memorization Stage:**
In Section 4.4, the table shows that the memorization stage plays a relatively minor role in the three-stage training process. However, the authors assert that this stage is beneficial for generalization, with further discussion supposedly found in Appendix F. There is, however, no such discussion present in Appendix F. I would recommend authors to do more experiments to validate the significance of the memorization stage and its impact on generalization.

**Hallucination Comparison:**
Section 5.4 contains statements that do not fully make sense. The claim that ToolGen experiences no hallucination is primarily due to the use of a constrained decoding strategy. Theoretically, it is impossible to encounter hallucination, as defined by the authors, under this constraint. This does not constitute a fair comparison with other frameworks. The authors should either provide results without this constraint and compare them or demonstrate why other frameworks are unable to implement a similar strategy.

**Performance Benchmarking:**
ToolGen is not currently the best-performing framework. Several baselines in the paper outperform ToolGen in various categories, as shown in Table 1 and Table 4. Notably, even the proposed but ultimately unused semantic tokenization approach outperforms the authors' current method, as shown in Table 2. This discrepancy confuses me.

**Questions:**

- As stated earlier, why don't take the semantic representation approach as your final approach as it performs better (5 wins / 4 loses in table 2) than atomic approach? Saving tokens might not be a sufficient reason. Chain-of-thought reasoning process should produce a lot more than what atomic approach can save here.

- In Table 1: "For ToolGen in the In-Domain setting, we allow the generation space to include all tokens, which is a more challenging scenario compared to other models." What if you don't include all tokens? Since ToolGen can't outperform IterFeedback in the In-Domain setting currently, will it be able to outperform it without this additonal challange?

- In section 4.4,  the authors are assessing the impact of different training stages, but the table shows "−memorization", "−retrieval training", "−constraining". If I understand correctly, constraining is an decoding strategy used in inference stage, while the third training stage "end-to-end agent-tuning" does not show up.

Please also pay attention to the questions and suggestions in the "Weaknesses" part.

---

> ### Author Response · Authors · 2024-11-19
> **Response to reviewer SiCu (part 1)**
>
> Thank you for your insightful and detailed comments. We noticed that several of your concerns asise from unclear descriptions or imprecise statements in our paper.
>
> We have proposed some revisions to address these concerns and reduce any potential misunderstandings. We hope the explanations provided below can alleviate your concerns.
>
> > Cost and Efficiency Claims
>
> Thank you for pointing this out. We will revise the statement "significantly less cost and higher efficiency" as "significantly less cost and higher efficiency at the same performance level".
>
> In terms of performance, it can be seen from Table 1 that ToolGen outperforms BM25, EmbSim, Re-Invoke,  and ToolRetriever methods by a large proportion (10%-60% improvement), and it achieves comparable results to IterFeedback.
>
> As for the cost and efficiency, IterFeedback employed a BERT model for retrieval and prompted an LLM (gpt-3.5-turbo in their paper) to iteratively provide feedback on retrieval results (3 rounds). In contrast, ToolGen performs the task in a single LLM inference step, generating only one token. This results in at least 3x savings in cost and latency.
>
> We will revise this statement and add the above detailed comparison.
>
> > Role of the Memorization Stage
>
> Thank you for your suggestion. From Table 3, we observe a performance drop of about 1-3% without the memorization stage. Regarding generalization, we noticed that Appendix F lacks a detailed discussion beyond the table. We have designed a new set of experiments to validate this aspect. We would greatly appreciate any feedback on our experimental design.
>
> First, we need to clarify the definition of generalization in this context.
>
> In our three-stage training process, the data in the first stage encompasses all tools. However, the training data in the second and third stages has limited tool coverage. This reflects a practical scenario where we may have access to the names and documents of more tools, but less coverage of the use cases of these tools in iterative training data. Generalization, in this case, refers to the model’s ability to correctly retrieve or use tools that were not included in the training data for stages 2 or 3. We believe that stage 1 plays a crucial role in achieving this, as without it, the retrieval or tool-using capabilities learned in later stages may not generalize to these unseen tools.
>
> We noticed that the retrieval training dataset contains approximately 500k samples, and the end-to-end training consists of 183k samples --- significantly more than the total number of tools (47k). This could result in most tools being seen during other training stages, which may affect the investigation of how memorization contributes to generalization.
>
> To validate how the memorization stage influences the generalization abilities of ToolGen, we designed the following experiments with controlled settings:
>
> First, we initialize two models --- one with tool memorization training and the other without. Then, we perform retrieval training and end-to-end training under controlled settings using subsets of the data:
> - 1/10 of the data
> - 1/2 of the data
> - A subset of data without the tools in the evaluation set.
>
> This controlled setting allows us to assess how significantly tool memorization contributes to generalization for both the general case and unseen tools.
>
> We are currently running these experiments and expect results within a few days. We would be delighted to hear any feedback on our experimental design in the meantime.

---

> > ### Author Response · Authors · 2024-11-19
> > **Response to reviewer SiCu (part 2)**
> >
> > > Hallucination Comparison
> >
> > We apologize for not clearly stating this in the original manuscript. Since this involves implementation details of constrained decoding for other methods and is not the main focus of this paper, we did not elaborate much on it. However, thank you for pointing this out. We agree that it is valuable to clarify this in the paper.
> >
> > Specifically, in the tool generation scenario, representing each tool with a single token is the most unbiased and efficient method for constrained decoding.
> >
> > For semantic encoding, the prevalent constrained beam search introduces bias toward tools with more subtokens after tokenization. Traditionally, beam search retrieves the top-k decoding sequences at each step, and the sequence probability is computed by multiplying the probabilities of each token (given previous tokens) in the sequence and then averaging by the token count. Consider the following example with two tools:
> > - toolA: get_music_from_us → [“get”, “music”, “from”, “usa”]
> > - toolB: get_music_from_spain_black_singer... (a long-tail name) → [“get”, “music”, “from”, “spain”, “black”, “singer”, …]
> >
> > After tokenization, the first three tokens are identical. For the fourth token, suppose "usa" has a probability of 0.7, and "spain" has a probability of 0.3. However, after "spain", toolB has a long tail of tokens with no alternatives, resulting in all subsequent tokens having a probability of 1.
> >
> > How should the best tool for decoding be determined in this scenario?
> >
> > Using the traditional method, toolB’s sequence probability increases as its unique number of tokens grows with each time step. This results in that tools with more unique tokens will have a higher probability to be retrieved.
> >
> > As shown in Figure 3, semantic tool name encoding tends to produce many long-name tools, and hence such bias becomes severe. However, there is no common solution to this type of bias. Note that this problem also exists in constrained natural language decoding. Still, because language candidates are typically very large, and constraints are usually associated with ban list of tokens or words, this issue is not usually considered.
> >
> > Based on the above observations, we made the hallucination comparison in paper, with a setting of non constrained beam search (to avoid length bias) for other encoding methods. For atomic encoding, hallucination and bias do not exist even with constrained decoding (not beam search) because each tool is represented by a single token, ensuring unbiased and deterministic decoding.
> >
> > Thank you again for pointing out this, we will add the above to the paper appendix.
> >
> > > Performance Benchmarking and reason for not choosing semantic indexing
> >
> > We have conducted two types of experiments: tool retrieval (Section 4), agent task (Section 5).
> >
> > For tool retrieval, we do not claim that ToolGen is the best. In Table 1, ToolGen achieves the best score in 4 out of 9 columns and ranks 2nd (very close to the best) in the remaining 5 columns. Therefore, we believe our claim of “comparable results to the best existing method” stands. In Table 2, we compare different tool encoding methods using the same training paradigm proposed in this paper and find that atomic encoding performs similarly to semantic encoding. However, we do not claim that atomic encoding is the best method for the tool retrieval task.
> >
> > For the agent task, from Table 4 and Table 5, we can clearly see that ToolGen achieves the highest scores across most metrics.
> >
> > If we only consider tool retrieval, the proposed semantic tokenization might perform well. However, after considering the agent task, the potential for tool hallucination (as mentioned above and in the paper), and the more realistic application of our methods in the agent scenario, we chose to use atomic encoding.
> >
> > If you have any suggestions to better illustrate these considerations and reduce confusion regarding the choice of encoding methods, please let us know. We will gladly consider incorporating clearer explanations.
> >
> >
> > >  For in-domain retrieval setting: Will ToolGen be able to outperform IterFeedback if we do not include tool tokens from other domains?
> >
> > As mentioned above, we input the query as a prompt and fetch the top-k tool tokens by ranking the next token probabilities as the top-k retrieval results. If we exclude irrelevant tokens from other domains, both precision and recall will increase or at least remain at the same level.
> >
> > In our previous experiments, we only fetch the top 5 tokens and did not save the rankings of all tool tokens.  To get empirical support for this argument, we are currently working on this and will report the results soon. Again, thank you for your question.

---

> > > ### Author Response · Authors · 2024-11-19
> > > **Response to reviewer SiCu (part 3)**
> > >
> > > > In section 4.4, the authors are assessing the impact of different training stages, but the table shows "−memorization", "−retrieval training", "−constraining".
> > >
> > > You are correct. We will revise our statement to “different training stages and constrained decoding.” We intentionally did not include end-to-end training in this section because we are only considering retrieval-related settings in this section.
> > >
> > > Finally, thank you again for your detailed feedback and suggestions. We believe incorporating the above clarifications will greatly enhance our paper.

---

> > > > ### Comment · Reviewer_SiCu · 2024-11-20
> > > >
> > > > Thank you for your detailed response. I believe that your ongoing experiments will address many of my concerns. Nevertheless, I think it may be beneficial to conduct additional experiments and garner statistical results to address performance and cost issues. While I agree that integrating the tool retrieval process into the LLM is more efficient, experimental data would make this claim more robust. (e.g. the speed and cost of each method)
> > > >
> > > > Furthermore, regarding your statement:
> > > >
> > > > > If we only consider tool retrieval, the proposed semantic tokenization might perform well. However, after considering the agent task, the potential for tool hallucination (as mentioned above and in the paper), and the more realistic application of our methods in the agent scenario, we chose to use atomic encoding.
> > > >
> > > > I would like to suggest that the authors conduct an experiment using semantic representation for agent tasks to substantiate this claim.
> > > >
> > > > Additionally, your assertion:
> > > >
> > > > > Specifically, in the tool generation scenario, representing each tool with a single token is the most unbiased and efficient method for constrained decoding.
> > > >
> > > > I recommend including this point in the main body of the paper, as it could help alleviate misunderstandings.
> > > >
> > > > There's still plenty of potential in this paper. Hope this helps!

---

> > > > > ### Author Response · Authors · 2024-11-23
> > > > >
> > > > > Thank you for your prompt reply, recognition of our experiments, and new suggestions!
> > > > > We have got all the results , including those promised in our previous response and those based on your new suggestion.
> > > > >
> > > > > > Performance and Cost Statistics
> > > > > The following table presents the speed and cost statistics of different baselines and ToolGen. Specifically, we report the mean and standard deviation of each method based on the execution of 100 queries.
> > > > >
> > > > > Note that for BM25, we use the open-source implementation from: https://pypi.org/project/rank-bm25/. The computation is performed on the CPU.
> > > > > For ToolRetrieval and ToolGen, we use a local machine with 1x A100 80GB GPU.
> > > > > For EmbSim, Re-Invoke, and IterFeedback, we follow the original implementation, which requires sending web requests to OpenAI, dominating the latency.
> > > > >
> > > > > From the table, we can see that any local implementation shows smaller latency than implementations requiring web requests. To make a fair comparison, we provide the required compute analysis. From the table, we can observe that BM25 requires the smallest compute resources but achieves the worst NDCG score. EmbSim and ToolRetriever have slightly higher compute requirements, with ToolRetriever achieving an average NDCG of ~72. IterFeedback achieves the best NDCG score, with ToolGen having a very small gap, while requiring more than 3x fewer compute resources.
> > > > >
> > > > > Method         | NDCG AVG | Actual Time (s) per query | Cost (USD) 100 queries | Required Compute
> > > > > ---------------|----------|---------------------------|------------------------|-------------------
> > > > > BM25           | 28.73    | 0.3778±0.0884            | $5.89\times10^{-3}$    | 1x string traversal + 1x matrix multiplication
> > > > > EmbSim         | 53.35    | 4.85±1.04                | $7.54\times10^{-2}$    | 1x LM encoding + 1x matrix multiplication
> > > > > Re-Invoke*     | 59.69    | 4.76±0.81                | $8.80\times10^{-2}$    | 1x LLM inference + 1x LLM encoding + 1x matrix multiplication
> > > > > ToolRetriever  | 71.96    | 0.0181±0.00048           | $2.01\times10^{-3}$    | 1x LM inference + 1x matrix multiplication
> > > > > IterFeedback*  | 89.51    | 9.90±0.53                | $5.47\times10^{-2}$    | 3x (LLM inference + LM encoding + matrix multiplication)
> > > > > ToolGen        | 89.29    | 0.0457±0.0019            | $5.08\times10^{-3}$    | 1x LLM inference for 1 token
> > > > >
> > > > > For cost estimation, we referred to the pricing on HuggingFace's website https://huggingface.co/pricing#endpoints . Specifically, we calculated the cost of processing 100 queries based hourly rate of CPU instances (8 vCPUs + 16GB Memory) for BM25, EmbSim, Re-Invoke, IterFeedback, and hourly rate of GPU instances (1 NVIDIA A100 with 80GB GPU Memory) for ToolRetriever and ToolGen. For methods that require GPT inference and encoding, we estimate the cost based on gpt-3.5-turbo and text-embedding-3-large, respectively.
> > > > >
> > > > >
> > > > > > Role of the Memorization Stage - Additional Results
> > > > >
> > > > > We have conducted additional experiments, and the results indicate that tool memorization significantly contributes to ToolGen's improved performance on tools that are unseen during retrieval training and agent tuning stage.
> > > > >
> > > > > We initialize two models: one without tool memorization and one with tool memorization. For the retrieval task evaluation, we train the model on I1 and test its performance on I2 and I3. There is very little tool overlap between different domains. From the table, we can see that the model without memorization training shows poor results on the retrieval task across different domains, while memorization greatly improves performance.
> > > > >
> > > > > | Setting          |       |  I2   |       |       |  I3   |       |
> > > > > | ---------------- | :---: | :---: | :---: | :---: | :---: | :---: |
> > > > > |                  | NDCG1 | NDCG3 | NDCG5 | NDCG1 | NDCG3 | NDCG5 |
> > > > > | w/o memorization | 11.28 | 16.72 | 19.37 | 12.00 | 15.07 | 18.15 |
> > > > > | w/ memorization  | 40.86 | 50.37 | 55.38 | 33.00 | 40.98 | 49.97 |

---

> > > > > > ### Author Response · Authors · 2024-11-23
> > > > > >
> > > > > > For the end-to-end task evaluation. As we proposed in previous response. We initialize two models: one without tool memorization and one with tool memorization. We then train the two models with 10%, 50% and full volume of retrieval data respectively, and finally train all of them with the full data of end-to-end agent-tuning.
> > > > > >
> > > > > > The results are shown in the following table. We can see that as the volume of retrieval data decreases, the importance of tool memorization increases (as validated by the increased gaps between the first row and second row from right to left).
> > > > > >
> > > > > > |                                       |               |   Pass Rate   |                |               |   Win Rate    |                |
> > > > > > | ------------------------------------- | :-----------: | :-----------: | :------------: | :-----------: | :-----------: | :------------: |
> > > > > > | Setting                               | 10% retrieval | 50% retrieval | 100% retrieval | 10% retrieval | 50% retrieval | 100% retrieval |
> > > > > > | w/o tool memorization                 |     36.60     |     48.67     |     51.70      |     26.68     |     37.32     |     39.32      |
> > > > > > | w/ tool memorization(+increased gaps) | 40.99 (+4.38) | 49.07 (+0.40) | 52.66 (+0.96)  | 34.23 (+7.55) | 37.60 (+0.28) | 39.53 (+0.21)  |
> > > > > >
> > > > > >
> > > > > >
> > > > > > For your new comments and suggestions.
> > > > > >
> > > > > > > I would like to suggest that the authors conduct an experiment using semantic representation for agent tasks to substantiate this claim.
> > > > > >
> > > > > > Thank you, in the table below, we directly compare the semantic and atomic representation for agent tasks. As you can see, the Atomic shows better performance than semantic encoding.
> > > > > >
> > > > > >
> > > > > > | Representation |          | SoPR     |          |       |          | SoWR     |          |       |
> > > > > > |----------------|----------|----------|----------|-------|----------|----------|----------|-------|
> > > > > > |                | I1 Inst. | I2 Inst. | I3 Inst. | AVG.  | I1 Inst. | I2 Inst. | I3 Inst. | AVG.  |
> > > > > > | Atomic         | 56.13    |  52.20   | 47.54    | 53.28 | 50.92    | 62.26    | 34.42    | 51.51 |
> > > > > > | Semantic       | 58.79    | 45.28    | 44.81    | 51.87 | 49.69    | 57.55    | 26.23    | 47.88 |
> > > > > >
> > > > > > > Include the point that representing each tool with a single token is the most unbiased and efficient method for constrained decoding in the main body of the paper, as it could help alleviate misunderstandings.
> > > > > >
> > > > > > Thank you for your suggestion! We believe this will improve our work significantly. As you suggested, we have added it to the paper and details of constrained decoding bias in Appendix E.
> > > > > >
> > > > > > We sincerely thank you for your questions and suggestions. We have added several new discussions to our paper accordingly. Hope these address your concerns.

---

> > > > > > > ### Comment · Reviewer_SiCu · 2024-11-25
> > > > > > >
> > > > > > > Thank you for your response. I will raise my score for the extra experiments.

---

### Official Review · Reviewer_pQUD · 2024-11-03

**Soundness:** 2
**Presentation:** 3
**Contribution:** 3
**Rating:** 5
**Confidence:** 3

**Summary:**

The paper introduces ToolGen, a novel framework that aims to enhance the interaction between large language models (LLMs) and external tools. ToolGen shifts away from traditional tool retrieval methods and instead integrates tool knowledge directly into the LLM's parameters. This is achieved by representing each tool as a unique token, allowing the LLM to generate tool calls and arguments seamlessly as part of its text generation process.

**Strengths:**

1. ToolGen elegantly combines tool retrieval and execution into a single generative process, eliminating the need for separate retrieval mechanisms. This streamlines tool interaction and enhances efficiency, particularly as the number of tools increases.
2. The use of constrained beam search during inference effectively restricts the output to valid tool tokens, significantly reducing the generation of nonexistent tools, a common issue in LLM-based agents.
3. ToolGen demonstrates its capacity to effectively handle a large repository of over 47,000 real-world tools, highlighting its scalability compared to existing methods that struggle with vast tool sets.
4. The authors spend good amount of effort comparing different indexing method, and the result is clear.

**Weaknesses:**

1. The advantage of ToolGen which combines tool retrieval and execution into a single generative process introduces limitation together with its efficiency. Since the tools are integrated into the system as tokens, the extension of new tools become inefficient. For every new tool/API, new token need to be added and the documentation finetuned into the model. Also, consider the case that when the tool/APIs get updated, the maintenance of all the tool/APIs, making sure they are up to date is a quite challenging task. On the contact, with a retriever would make adding and maintaining tool/APIs very easy.

**Questions:**

1. The paper highlights the efficiency of atomic indexing due to each tool being represented by a single token. However, as the number of tools grows significantly larger, could this approach lead to vocabulary explosion issues?
2. In the current setting, the vocabulary consists of 128K regular token and 47K tool/API tokens already, how does it affect the performance of LLM itself?
3. How does the performance of ToolGen vary with the size and complexity of the underlying LLM? Would a larger model lead to better generalization ability?

---

> ### Author Response · Authors · 2024-11-19
> **Response to Reviewer pQUD Part 1**
>
> Thank you for your thorough review and valuable feedback on our work. We address your concerns as follows:
> > Inefficiency of tools/APIs maintenance and adding new tool/APIs
>
> Thank you for pointing out this problem when tool set changes. We will add the following discussion about this limitation in the revised version.
>
> To address this problem,  during inference, ToolGen does not only generate the proper tool, but also fetch the documentation for that tool. Therefore, for minor changes that the tools usage scenarios keep the same (e.g. parameter changes), ToolGen can still generate the tool and rely on fetched documentation to do further tasks.
>
> For vast changes that the usage scenarios are different or adding new tools, we admit that ToolGen is not able to utilize these tools. However, this inefficiency exists and is persistent for generative retrieval systems [1,2,3]. For changes in existing tools, adaptation can be done with knowledge editing. For adding new tools, current research uses continual training and constrained optimization [3,4], which we believe could also be applied to ToolGen to alleviate the above challenges.
> Despite the fact that ToolGen is less efficient at adopting new tools, its unified design leads to unique advantages such as end-to-end training and an easy integration with Chain-of-Thought, Reinforcement Learning with Human Feedback, and inference scaling.
>  We leave the problem of maintaining and adding tools to future work.
>
> > As the number of tools grow significantly larger, will there be a vocabulary explosion issue?
>
> We think this could happen to our model. However, to the best of our knowledge, ToolBench, the dataset in our experiments, is the largest tool dataset. We demonstrate that ToolGen works efficiently and effectively with such scale tools. To mitigate the possible issue for an extremely larger tool set (10x or 100x larger than the existing setting), a possible solution is to adopt other encoding methods to encode tools, which we have discussed in our paper. For example, use hierarchical or semantic encoding methods.
>
> We believe that many techniques developed at certain stages need to be revised or adapted to accommodate large-scale data as time progresses. Recently, we have faced the practical challenge of working with large-scale tool-learning agents (even thousands of tools). While our method mitigates these issues and we anticipate it will continue to solve problems in the coming years.
>
> > How does adding tool/API tokens affect the performance of LLm itself
>
> For this point, we need to clarify that a robust fine-tuned agent with automatic tool retrieval and execution abilities is different from a general-purpose LLM. This mode shift makes it challenging to perform other tasks as the original model would. That’s why in previous agent-tuning or tool-learning papers, including but not limited to Toolformer [5] and ToolKenGPT [6], such results are not presented. We followed previous work and only reported the results over the widely used benchmark datasets.
>
> We also think it is valuable to compare other capability of LLMs before and after tool token incorporation. We are currently running evaluations, it is costly and time-consuming and we expect to get results in two days. We hope that even less favorable results will not diminish the assessment of this work. Thank you for your understanding.

---

> ### Author Response · Authors · 2024-11-19
> **Response to Reviewer pQUD Part 2**
>
> > Performance and generalization of ToolGen with varying sizes.
>
> To conduct experiments of ToolGen in different sizes under the premise of controlling variables, Llama 3 models are not suitable as we can only use the 8B model and the 70B is too large for us. Therefore, we select Qwen2.5 with sizes of 1.5B, 3B, 7B, and 14B. The following table shows the experimental results of ToolGen with different sizes. Generally, models with larger sizes achieve better performance in tool retrieval and agent tasks. When the model size reaches 7B, the performance tends to plateau. And scaling it to 14B does not necessarily improve performance.
>
> For generalization evaluation, we are currently still running the experiments and will update the results as soon as possible.
>
> - Tool Retrieval Evaluation:
>
> | Model                |       |  I1   |       |       |  I2   |       |       |  I3   |       |
> | :------------------- | :---: | :---: | :---: | :---: | :---: | :---: | :---: | :---: | ----- |
> |                      | NDCG1 | NDCG3 | NDCG5 | NDCG1 | NDCG3 | NDCG5 | NDCG1 | NDCG3 | NDCG5 |
> | ToolGen-Qwen2.5-1.5B | 87.67 | 89.36 | 91.92 | 84.97 | 84.63 | 88.20 | 70.00 | 71.16 | 81.26 |
> | ToolGen-Qwen2.5-3B   | 90.33 | 90.44 | 93.12 | 85.72 | 84.75 | 88.76 | 84.00 | 79.88 | 87.20 |
> | ToolGen-Qwen2.5-7B   | 91.67 | 92.35 | 94.38 | 89.23 | 88.15 | 91.71 | 80.00 | 82.52 | 86.32 |
> | ToolGen-Qwen2.5-14B  | 90.83 | 91.55 | 93.87 | 89.23 | 88.51 | 91.33 | 82.00 | 81.86 | 88.64 |
>
> - End-to-end Evaluation
>
> | Model                 |          |   SoPR   |         |       |          |   SoWR   |         |       |
> | :-------------------- | :------: | :------: | :-----: | :---: | :------: | :------: | :-----: | :---: |
> |                       | I1-Inst. | I2-Inst. | I3-Inst | AVG.  | I1-Inst. | I2-Inst. | I3-Inst | AVG.  |
> | ToolGen-Qwen-2.5-1.5B |  56.85   |  48.90   |  47.54  | 52.58 |  44.17   |  50.94   |  40.98  | 45.75 |
> | ToolGen-Qwen-2.5-3B   |  60.12   |  45.75   |  44.26  | 52.57 |  53.37   |  50.94   |  36.07  | 49.39 |
> | ToolGen-Qwen-2.5-7B   |  58.79   |  56.76   |  56.83  | 57.78 |  51.53   |  63.21   |  49.18  | 54.85 |
> | ToolGen-Qwen-2.5-14B  |  53.48   |  52.20   |  74.86  | 57.02 |  42.94   |  55.66   |  59.02  | 50.00 |
>
> [1] Sun, Weiwei, et al. "Learning to tokenize for generative retrieval." Advances in Neural Information Processing Systems 36 (2024).
>
> [2] Chen, Jiangui, et al. "Continual learning for generative retrieval over dynamic corpora." Proceedings of the 32nd ACM International Conference on Information and Knowledge Management. 2023.
>
> [3] Mehta, Sanket Vaibhav, et al. "DSI++: Updating Transformer Memory with New Documents." Proceedings of the 2023 Conference on Empirical Methods in Natural Language Processing. 2023.
>
> [4] Kishore, Varsha, et al. "Incdsi: incrementally updatable document retrieval." International Conference on Machine Learning. PMLR, 2023.
>
> [5] Schick, Timo, et al. "Toolformer: Language models can teach themselves to use tools." Advances in Neural Information Processing Systems 36 (2024).
>
> [6] Hao, Shibo, et al. "Toolkengpt: Augmenting frozen language models with massive tools via tool embeddings." Advances in neural information processing systems 36 (2024).

---

> > ### Comment · Reviewer_pQUD · 2024-11-20
> > **Thanks for the reply.**
> >
> > Thanks for the reply, will keep my rating.

---

> > > ### Author Response · Authors · 2024-11-23
> > > **Thanks for your response.**
> > >
> > > Thanks for your response.
> > >
> > > Regarding the additional experimental results on ToolGen's general capabilities as we promised before, we have conducted experiments, and the results indicate that ToolGen performs poorly on general tasks, but fine-tuning on data with mixed general instructions (i.e., ToolGen-Instruct) not only prevents a decline in general performance but also enhances its effectiveness in tool learning. Please refer to our **Response to Reviewer ieJ7 (Part 3)** https://openreview.net/forum?id=XLMAMmowdY&noteId=RKO4jmNCyP.

---

### Official Review · Reviewer_ieJ7 · 2024-11-04

**Soundness:** 2
**Presentation:** 2
**Contribution:** 2
**Rating:** 5
**Confidence:** 5

**Summary:**

This paper proposes ToolGen, a finetuned LLMs that can use various tools during the conversations with the users. ToolGen incorporates new 47K tokens for tools into the Llama-3-8B. Through the tool virtualization, tool memorization, retrieval training, and end-to-end agent tuning, ToolGen correctly select the right tools in the context on  the ToolBench evaluation, achieving better performance than retrieval and end-to-end baselines.

**Strengths:**

- [S1] ToolGen outperform or achieves competitive performance among retrieval and end-to-end baselines on ToolBench.
- [S2] ToolGen can natively invoke 47K tools following the context.

**Weaknesses:**

- [W1] The technical novelty is limited. Using special tokens for tools and incorporating them into the original vocabularies are widely-known approach (e.g. Toolformer: https://arxiv.org/abs/2302.04761, ToolkenGPT: https://arxiv.org/abs/2305.11554). The contribution of this paper is scaling this up to 47K tools, but it's very straight forward and I'm not confident if the ICLR community would be interested in it.
- [W2] Releted to [W1], the results of ToolLlama-3 in Section 5 is unclear to me. Why ToolLlama-3 is not as good as ToolGen, even using the same data and models? Could you clarify the difference between two?
- [W3] How about retrieving tools by LLMs itself (without retriever)? For instance, if we use the LLM with millions of extremely long context (such as Gemini), we may not need to rely on neural retrievers. It would be interesting to include long-contect LLMs as a baseline.
- [W4] It would be important to compare other capability of LLMs before/after tool token incorporation. Current draft seems to lack the analysis after combining 47K tokens to LLMs.
- [W5] In Figure1, I didn't understand the distinction between ToolGen's Retrieval Task and Agent Task. Both have the input "I want some popular video games." How are they differentiated?

**Questions:**

See the Weaknesses above.

---

> ### Author Response · Authors · 2024-11-19
> **Response to Reviewer ieJ7 Part 1**
>
> Thank you for insightful feedback. We have tried to resolve your concerns as follows:
>
> > W1: Novelty of our approach and its relationship to ToolKenGPT and other methods.
>
> We apologize for not making our contributions clear enough. Hopefully, the following explanation, with specific examples, will make it clearer (we will add the following parts to the paper).
>
> Regarding the use of special tokens, we acknowledge that previous work already employed vocabulary expansion for tool learning. However, the main difference between our work and others is: as we mentioned in Section 2 Line 147, previous studies primarily demonstrate that through SFT (in Toolformer) or adding new tool tokens with pre-computed embeddings (in ToolKenGPT), LLMs can learn to use a very small number of tools. However, in real-world tool-calling (agent) scenarios, previous methods require listing available tools in the prompt, which greatly limits their practical use. See example 1, 2, and 3 below.
>
> Scaling up tool learning to 47k in a practical agent scenario seems to be straightforward, but actually, it's not. In this scaling process, we encountered several issues. Although we did not explicitly enumerate these issues, we described our solutions in the paper. Specifically:
> (1) What type of training is needed for an LLM to learn to use a large number of tools?
> Previous methods that simply expand the vocabulary with pre-computed embeddings are ineffective for this purpose. Key issues include: (i) it is hard to capture nuanced difference in pre-computed tool representation; and (ii) it is almost impossible to put such large number of tools into a prompt. Therefore, we proposed a new paradigm that consists of: tool memorization for tool representation learning, retrieval training for tool-query association learning, and end-to-end training for better tool calling and integration to LLM parameters. We explained the necessity of each step into detail in Section 3 and demonstrated that with experiments in Sections 4,5.
>
> (2) In addition to the **atomic encoding** approach, in this paper, we have gained inspiration from previous work and tried three other tool enconding methods: the **semantic encoding** approach that has been widely used in previous tool learning work which does not require extra tokens; the **numeric encoding** approach which has been used in generative document retrieval tasks but is not well studied for tool learning; and the **hierarchical encoding** approach also exist in [1, 2]. With these four different tool encoding approaches, it is not clear which one works better for tool learning, especially in real-world tool-calling scenarios that incorporate a large scale of tools. We are the first to systematically explore and compare these encoding methods of tool token representations, which we believe will inspire further tool retrieval and tool learning research.
>
> (3) How can we alleviate tool name hallucination?
> When the number of tools is limited and each tool is sufficiently trained (see Examples 1, 2, and 3 below), we do not need to consider tool hallucination. However, as the tool count increases, hallucination becomes a significant issue, especially for tools that only with documents but not presented in agent training data. We propose a constrained decoding method to address this issue, and conduct experiments to demonstrate its effectiveness.
>
> ```
> Example 1 (from ToolkenGPT):
> Answer the following questions with <add>, <subtract>, <multiply>,
> <divide>, <power>, <sqrt>, <log>, <lcm>, <gcd>, <ln>, <choose>,
> <remainder>, and <permutate>:
> Question: A coin is tossed 8 times, what is the probability of getting
> exactly 7 heads?
>
> Example 2 (from ToolkenGPT):
> I am a household robot and I can take actions from ’[FIND]’, ’[SIT]’,
> ’[SWITCHON]’, ’[TURNTO]’, ’[LOOKAT]’, ’[TYPE]’, ’[WALK]’, ’[LIE]’,
> ’[GRAB]’, ’[READ]’, ’[WATCH]’, ’[POINTAT]’, ’[TOUCH]’, ’[SWITCHOFF]’,
> ’[OPEN]’, ’[PUSH]’, ’[PUTOBJBACK]’, ’[CLOSE]’, ’[DRINK]’, ...
>
> Example 3 (from Toolformer):
> Your task is to complete a given piece of text by using a Machine Translation API.
> You can do so by writing "[MT(text)]" where text is the text to be translated into English.
> Here are some examples:
> Input: He has published one book: O homem suprimido (“The Supressed Man”)
> Output: He has published one book: O homem suprimido [MT(O homem suprimido)]
> (“The Supressed Man”)
>
> Example 4 (ours, the same example is provided in the paper appendix)
> You are an AutoGPT, capable of utilizing numerous tools and functions to complete the given task.
> 1.First, I will provide you with the task description, and your task will commence.
> 2.At each step, you need to determine the next course of action by generating an action.
> ... (no specific tool or API mentioned in instruction)
> Could you please fetch the addresses for the postcode 'PL11DN'? I would like to know the number of items found, the district, ward, county, country, and geocode details (eastings, northings, latitude, and longitude).
> ```

---

> ### Author Response · Authors · 2024-11-19
> **Response to Reviewer ieJ7 Part 2**
>
> > [W2] Why ToolLlama-3 is not as good as ToolGen, even using the same data and models? Could you clarify the difference between two?
>
> There are three main differences.
> 1. ToolGen contains more tool knowledge. ToolGen includes two additional training stages: tool memorization and retrieval training, that allow it to develop a deeper understanding of tools.
> 2. ToolGen leverages knowledge from retrieval training stage. We notice that there is a similarity between retrieval training and end-to-end training: both take a user query as input, and generate relevant tools. Compared to ToolLlama, ToolGen has an extra retrieval training stage, making it better associate queries with relevant tools.
> 3. Ability and possibility to explore alternative tools. We noticed that the annotated “ground truth” tools are not the only ones capable of solving a given task; they are just one set of possible tools annotated by previous researchers. In our experiments, we observed that some tools outside the annotated ground truth tools can also solve the tasks, while ToolGen can utilize them by iteratively exploring all possible tools, ToolLlama can only explore a fixed number of tools (usually top 5 retrieved). Note that we use the same evaluation criteria and evaluator in all experiments, consistent with previous study.
>
> In summary, ToolGen’s memorization stage, combined with retrieval training, enables the model to leverage memorized tool knowledge and generalize to non-annotated tools. This is challenging for other methods that rely on separate models for retrieval and agent task.
>
>
> > [W3] How about retrieving tools by LLMs itself without retriever (long-context baseline)?
>
> Thank you for reminding us! It's a good idea and we believe it will enhance our paper. To get a long-context model baseline, we selected GPT-4o-mini and GPT-4o as baseline models, which have a context length of 128k. However, since there are about 47k tools, which can not be fitted into the context, we select 2k of them (ground truth tools are included) to form a long prompt. Results are shown in the following table, we can see, although with a much easier setting (2k compared to 47k), GPT-4o-mini and GPT-4o have a huge gap to ToolGen.  This showcases the ineffectiveness of putting all tools into prompts. At the same time,  we noticed that it has a much higher cost: Around 20 USD for testing only 100 examples for GPT-4o.
>
> | Model       |       | I1    |       |       |  I2   |       |       |  I3   |       |
> | ----------- | :---: | ----- | :---: | :---: | :---: | :---: | :---: | :---: | :---: |
> |             | NDCG1 | NDCG3 | NDCG5 | NDCG1 | NDCG3 | NDCG5 | NDCG1 | NDCG3 | NDCG5 |
> | GPT-4o-mini | 27.52 | 40.49 | 45.34 | 20.43 | 33.44 | 38.49 | 19.63 | 33.86 | 41.81 |
> | GPT-4o      | 32.22 | 42.87 | 52.14 | 25.39 | 33.91 | 46.07 | 25.11 | 32.57 | 44.03 |
>
>
> > [W4] Compare other capability of LLMs before/after tool token incorporation.
>
> For this point, we need to clarify that a robust fine-tuned agent with automatic tool retrieval and execution abilities is different from a general-purpose LLM. This mode shift makes it challenging to perform other tasks as the original model would. That’s why in previous agent-tuning or tool-learning papers, including but not limited to Toolformer and ToolKenGPT, such results are not presented. We followed previous work and only reported the results over the widely used benchmark datasets.
> We also think it is valuable to compare other capability of LLMs before and after tool token incorporation. We are currently running evaluations, it is costly and time-consuming and we expect to get results in two days. We hope that even less favorable results will not diminish the assessment of this work. Thank you for your understanding.
>
> > [W5] In Figure1, The distinction between ToolGen's Retrieval Task and Agent Task.
>
> The key differences are that in the retrieval task, after we input the query and prompt, we fetch the logits of the generated next token and get the top-k tool tokens as the results. In contrast, in the Agent task, we allow the model to perform generation, which usually involves generating one tool with the highest probability and then generating its parameters.
>
> [1] Wang, Yujing, et al. "A neural corpus indexer for document retrieval." Advances in Neural Information Processing Systems 35 (2022): 25600-25614.
>
> [2] Li, Yongqi, et al. "Generative cross-modal retrieval: Memorizing images in multimodal language models for retrieval and beyond." arXiv preprint arXiv:2402.10805 (2024).

---

> > ### Author Response · Authors · 2024-11-23
> > **Response to Reviewer ieJ7 (Part 3)**
> >
> > We appreciate your patience regarding the experimental results. While we initially anticipated completing the experiments within two days, the process took slightly longer than expected. We apologize for the delay and hope the results meet your expectations.
> >
> > To test the model's general ability, we used 6 widely used LLM evaluation benchmarks, including ARC_easy, ARC_challenge, Commonsense QA, Hellaswag, Winograde, and GSM8K, all with 3-shot in-context examples.
> >
> > As expected, ToolGen’s general capability is limited. The original Llama 3 got an average score of 63.76, while ToolGen obtained almost random results with a score of 26.94, showing a significant gap. However, inspired by your suggestion, ToolGen fine-tuned on data with mixed general instructions has shown significant improvement, resulting in a 67.4 average score.
> >
> > Thanks for your suggestion, we have been thinking about how to keep the model's general ability after Tool-specific task training. To this end, we incorporated general instruction-following data (OpenHermes-2.5, https://huggingface.co/datasets/teknium/OpenHermes-2.5) into each training stage of ToolGen. We used a 50:50 ratio of our Tool data and instruction-following data, calling the resulting model ToolGen-Instruct, indicating that it can follow general instructions.
> >
> > In our evaluation, we obtained very promising results with an average score of 67.40, even higher than the original Llama-3, and a small gap compared to Llama-3-Instruct, which has undergone SFT and RLHF.
> >
> > | Task            | ARC_challenge | ARC_easy | Commonsense QA | Hellaswag | Winograde | GSM8K | AVG.  |
> > |------------------|---------------|----------|----------------|-----------|-----------|-------|-------|
> > | Llama-3         | 50.34         | 80.09    | 69.37          | 60.13     | 73.32     | 49.28 | 63.76 |
> > | ToolGen         | 20.14         | 33.88    | 19.66          | 31.61     | 54.62     | 1.74  | 26.94 |
> > | Llama-3-Instruct| 57.16         | 85.31    | 78.05          | 58.69     | 76.24     | 74.45 | 71.65 |
> > | ToolGen-Instruct| 51.62         | 82.49    | 78.79          | 56.33     | 73.16     | 62.02 | 67.40 |
> >
> > To check ToolGen-Instruct’s tool learning results, we conducted the end-to-end evaluation. Surprisingly, we found that its average performance improved by 3-5 percentage points compared to ToolGen. We did a manual comparison and found that ToolGen-Instruct performs better on final output summarization and provides more positive responses to the user after tool calling. From this observation, we concluded that general-purpose training can help ToolGen provide more helpful responses.
> > | Model            |          |   SoPR   |          |       |          |   SoWR   |          |       |
> > | ---------------- | :------: | :------: | :------: | :---: | :------: | :------: | :------: | :---: |
> > |                  | I1 Inst. | I2 Inst. | I3 Inst. | AVG.  | I1 Inst. | I2 Inst. | I3 Inst. | AVG.  |
> > | ToolGen          |  56.13   |  52.20   |  47.54   | 53.28 |  50.92   |  62.26   |  34.42   | 51.54 |
> > | ToolGen-Instruct |  63.09   |  55.03   |  54.10   | 58.84 |  51.53   |  60.38   |  50.81   | 54.24 |
> >
> > We greatly appreciate your feedback, we have incorporated the example comparison of ToolGen, ToolkenGPT and Toolformer into our paper Appendix A, and incorporated the general capability experiment and findings into appendix F (we will put these to the main body of paper later).

---

> ### Comment · Reviewer_ieJ7 · 2024-11-27
>
> Thank you authors for the detailed response and additional experiments. I respect the effort from the authors even in a limited timeline. Also, I really sorry for the late reply due to my personal matter.
>
> I still have a concern on the technical novelty, but the position of this work became clear. My other concern is whether broad ICLR community can have an interest in this work. This work increased the number of tool tokens up to 47K, but this is not the technical upper bound, rather the bound from the current data resources. We may see the doubled/tripled/N-times tool-token paper in next year, next venue, and future venues. This is a kind of applied research, and I agree that some have an interest on this for the product or services. But at the same time, this may not directly addresses the technical challenges, which may not be aligned with the research community.
>
> For the assessment, I would maintain the negative stance in the borderline.

---

### Author Response · Authors · 2024-12-03
**General Response**

We thank all reviewers for their detailed reviews and insightful feedback. Their comments will help to make our work more solid and comprehensive. Below is the summary of new results and updates made based on feedback from reviewers:

- We analyzed the latency and cost for ToolGen and different baselines, showing that ToolGen is significantly more effective compared to same performance-level baselines.
- We added discussions on tool maintenance and tool set change in Appendix B.
- We discussed how the general capabilities change in Appendix F.
- We added long-context LLM baseline in experiments.
- We compared atomic and semantic encoding to show atomic encoding's superiority in efficiency, hallucination and performance.
- We added and discussed how ToolGen's performance and generalization changes as the size changes in Appendix H.
- We added more comprehensive experiments and discussion to emphasize the role of tool memorization stage.
- We added more related work.

**Further clarification about our contributions**

First, we express our gratefulness to Revewer A87g, who has given highly positive and encouraging feedback to our work. We really appreciate Reviewer A87g's support for our work and we discuss more about our contribution and novelty here, which is also still concerned by Reviewer ieJ7:
- While the scalability to 47K tools is a notable aspect, we emphasize that ToolGen is not just about scaling but about redefining tool retrieval and usage as a unified generative task, which is also stated by Reviewer A87g. This shift addresses practical challenges, such as context length limitations and tool hallucinations, with a systematically designed pipeline integrating memorization, retrieval, and end-to-end tuning.
- Regarding scalability, while further expansions in tool tokens might appear as a natural progression, the methodology and infrastructure we propose (e.g., constrained decoding, multi-stage training) offer **first** foundational insights that extend beyond mere scaling and are applicable to future challenges in integrating even broader sets of tools.
- Regarding the interests of the community, the ToolGen framework is not confined to Tool Learning itself. In fact, it contributes to broad research topics.  For example, in areas which can be enhanced significantly by incorporating tools, such as reasoning, question-answering, information retrieval, RLHF, optimization is often hindered by the separation between tool selection and tool usage. ToolGen offers an option to internalize related tools into their parameters, therefore enabling end-to-end approaches for them to integrate massive and complex tools during training; In In-Context Learning, ToolGen makes it possible to do few-shot generation with a large set of available tools.

Therefore, we believe that ToolGen not only addresses existing challenges but also contributes to other research directions and inspires future works. We thank all reviewers' efforts to help improve our work and we are committed to incorporating further insights and feedback to enhance the impact and rigor of our contributions.

---

### Meta-Review · Area_Chair_JS9Y · 2024-12-20

**Metareview:**

The paper introduces ToolGen, a novel framework that integrates tool knowledge directly into the parameters of a Large Language Model (LLM) by representing each tool as a unique token.  This allows the LLM to generate tool calls and arguments seamlessly, eliminating the need for separate tool retrieval systems.  ToolGen is trained in three stages: tool memorization, retrieval training, and end-to-end agent tuning.  The authors claim this framework enhances the interaction between LLMs and external tools, particularly with large tool sets.

Strengths: The paper proposes a novel approach to tool retrieval and execution by representing tools as virtual tokens, streamlining the process.  The use of constrained beam search effectively reduces the generation of nonexistent tools.  The authors demonstrate the scalability of ToolGen by incorporating over 47,000 real-world tools, and how it can more easily deal with context length limitations and tool hallucinations.

Weaknesses: (1) Dealing with new tools at test-time or major updates in tool APIs would necessitate retraining, which has been added as a discussion in appendix.  (2) Degradation in general instruction following performance of LLMs when fine-tuning heavily on tool use trace, which would be critical for practical deployment. (3) Post-hoc tool use is common for deployment due to latency introduced by tool calls during generation, which might need to carefully handled with ToolGen.

Reasons for Acceptance/Rejection: The paper present a novel perspective on tool use -- generative tool calling - which can tap into the generation abilities of LLMs. Moreover, the authors have addressed the reviewers' concerns by providing additional explanations and committing to further experiments. The revised version of the paper, incorporating the planned improvements, is expected to be a valuable contribution to ICLR.

Minor suggestion: Work in similar spirit to ToolGen has been going on in generative verifiers / reward models (GenRM, Cloud) which would be worth discussing.

**Additional Comments On Reviewer Discussion:**

Reviewers expressed concerns about the paper's claims of reduced cost and efficiency, the role of the memorization stage, the fairness of hallucination comparisons, ToolGen's performance compared to other frameworks, and the novelty and clarity of the results.  The authors responded by promising to provide data and experiments supporting the cost and efficiency claims, clarifying the memorization stage and outlining further experiments, and explaining the hallucination comparison in detail.  They also acknowledged that ToolGen isn't always the best-performing framework but emphasized its overall performance in the agent task, and they provided clearer explanations and promised additional experiments to address the concerns about novelty and result clarity.  These responses, particularly the planned experiments and clarifications regarding the memorization stage, hallucination comparison, and ToolGen's overall performance, substantially changed my view to accept the paper.

---

### Decision · Program_Chairs · 2025-01-22

Accept (Poster)